

# 15th century climate in the Czech Lands and its Central European context

Rudolf Brázdil[1,2], Petr Dobrovolný[1,2], Max Carl Arne Torbenson[2,3], Lukáš Dolák[1,2], Kateřina Chromá[2]

[1]Institute of Geography, Masaryk University, Brno, Czech Republic
[2]Global Change Research Institute, Czech Academy of Sciences, Brno, Czech Republic
[3]Department of Geography, Johannes Gutenberg University, Mainz, Germany

*Correspondence to*: Rudolf Brázdil (brazdil@sci.muni.cz)

**Abstract.** Information concerning the weather and related phenomena in the Czech Lands (recently the Czech Republic) in the 15th century may be derived from the Old Czech Annals, chronicles, letters and from the surviving accountancy records of the town of Louny. Secondary sources are only of limited use. Critical evaluation of data from such reported sources, originating almost exclusively from Bohemia (the western part of the Czech Lands), facilitates a degree of description of the course of weather and related phenomena on an annual basis, but not for all years. Records for the 1400s and 1410s are particularly poor, while better evidence appears in the 1430s and 1450s. In order to interpret the temperature and precipitation character of this century, a 3-degree scale for months and a 7-degree scale for seasons were deployed to create series of temperature and precipitation indices. These indices are relatively more frequent for winter and summer, while far fewer indices can be derived for spring and autumn. Despite their incompleteness, Czech temperature and precipitation indices accurately reflect the occurrence of significant extremes and climate anomalies in Central Europe during the 15th century. Comparison with existing proxy reconstructions and paleo-reanalysis shows that the 15th-century Czech indices provide unique information, especially about December–February temperature variability, which is not easy to obtain from the study of natural proxies.

## 1 Introduction

Documentary evidence constitutes an important source of information for the study of climatic patterns, their anomalies and hydrometeorological extremes (HMEs) in the pre-instrumental period. Europe is a region in which extremely rich documentary evidence exists, covering many centuries (Brázdil et al., 2005, 2010). Of the plethora of historical-climatological studies addressing documentary data, a number of monographs are especially worthy of note (e.g., White et al., 2018; Pribyl and Kiss, 2020; Pfister and Wanner, 2021).

European 15th-century climate has been investigated as part of several studies covering longer timespans (e.g., Lyakhov, 1984; Alexandre, 1987; van Engelen et al., 2001; Shabalova and van Engelen, 2003; Rohr, 2007; Glaser, 2008; Telelis, 2008; Glaser and Riemann, 2009; Wetter and Pfister, 2011; Pribyl et al., 2012; De Kraker, 2013; Litzenburger, 2015; Labbé et al., 2019; Retsö and Söderberg, 2020; Przybylak et al., 2023; Pfister et al., 2024). Some of these contributions focused on



specific climate variables and confined regions. For example, Camenisch (2015) reconstructed temperatures and precipitation of the Burgundian Low Countries during the 15th century. Kiss and Nikolić (2015) and Kiss (2017) analysed several years with droughts and low water levels in late-medieval Hungary. Camenisch et al. (2016) analysed a cold period that took place in the 1430s in Europe. Camenisch et al. (2020) described extreme heat and drought, together with their

impacts for 1473, within the context of the early 1470s in Europe. Kiss (2020) analysed weather and weather-related natural hazards in medieval Hungary for 1401–1450 CE.

The Czech Lands (recently the Czech Republic) are among the European regions rich in documentary evidence, particularly from the 16th century. This has enabled detailed reconstructions of air temperature (Dobrovolný et al., 2010; Možný et al., 2012, 2016a), precipitation (Dobrovolný et al., 2015) and drought (Brázdil et al., 2016; Možný et al., 2016b) for the past 500

years. Brázdil et al. (2022a) used these reconstructions to analyse their forcings and European context over the past 500 years. However, surviving Czech documentary evidence before the 16th century is sparser, permitting description of only some weather/climatic patterns and HMEs in certain years (Brázdil and Kotyza, 1995, 1997), which were used, for example, by Brázdil et al. (2017a) to analyse severe famines in the 1280s, 1310s, and the early 1430s in the Czech Lands in relation to weather and climate conditions.

The aim of this contribution is to address research gaps concerning the 15th century in the Czech Lands by presenting the existing knowledge related to weather/climate and HMEs from available documentary evidence. The analysis concentrates on climate variability expressed by temperature and precipitation indices, documented HMEs, and comparison of these results with other climate reconstructions and data sources from Central Europe.

## 2 The Czech Lands in the 15th century

The 15th century in the Czech Lands lay in thrall to religious upheaval (the Hussite revolution), wars, frequent changes of rulers, political and economic isolation in Europe and a sharp deterioration in the cohesion of society, trends dating back as far as the 1380s. Economic recession, a proliferation of lawlessness (brigandage and widespread pillage) and frequent plague epidemics combined to cause deep societal crisis, radicalization of certain sections of the population, even the emergence of anarchy, particularly during the reign of Wenceslas IV (1378–1419). The quest for relief from such crises resulted in

collective assignment of blame to the Catholic Church, perceived as corrupt and inept. Inevitably enough, many clerics found widespread sympathy for their proselytizing sermons, preaching radical reform and a return to "original" Christian values (Čornej, 2003). However, the teachings of the most renowned and successful of them, theologian and philosopher Jan Hus, burned for heresy in 1415 in Konstanz (Germany), far from solving the problem, merely fanned the flames of dissent. Radicalization, societal fragmentation and polarization peaked in the Hussite Revolution of 1419 and the origin of the

Hussite movement, forerunner of Protestantism in general. The Hussites felt called to bring Christians back to their duties and to spread their ideology beyond the Czech Lands. However, Catholic Europe declared the Hussites heretics and five papal crusades against them were organised between 1420 and 1431; none of them was successful. However, the Hussite



movement was far from uniformly supported in the Czech Kingdom, with opponents in Upper and Lower Lusatia, and Silesia, as well as among the Czech nobility, largely Catholic. The resulting Hussite wars lasted from 1419 to 1436, leading

not only to the total isolation of the Czech Lands, but also to acute deterioration in education levels, widespread iconoclasm, the utter extirpation of many settlements, population decline, and societal divisions based on perceived nationhood and faith, even among "native" Czechs (Čornej, 2000).

Once the Hussite wars were over, their effects particularly devastating to rural areas, society continued to be split between Hussites-Calixtines and Catholics. While the Catholic Church lost almost all its property in rural Bohemia, the nobility and

occupants of towns became politically and economically stronger. Catholic kings (Sigismund of Luxembourg 1436–1437; Albrecht Hapsburg 1437–1439) soon took the Czech throne and, after a following "interregnum", King Ladislaus Posthumous (1453–1457). However, the whole of this period was riddled with religious controversy, which only ended with the first truly-elected Czech king, a Hussite known as Jiří (George) of Poděbrady (reigned 1458–1471) (Čornej, 2003).

George's reign was characterised by religious tolerance, political stability and economic recovery, although business contact

with the rest of Europe was not permitted; Bohemia was declared "heretical" by the Vatican (Calixtines made up *c*. 70% of the population) and the Czech king himself was considered a "Hussite heretic". After a dispute with Pope Pius II, he was deposed by the Catholic Church in 1466 and cast into anathema. When the Catholic estates declared Matthias I Corvinus as the Czech king in 1469, the country become a diarchy and a period of wars and economic recession began (Čechura, 2010).

Economic deterioration and general exhaustion of the country's resources, together with a sharp rise in crime, led in 1479 to

reconciliation between Matthias I Corvinus and the new Czech king Vladislaus II Jagiellon (1471–1526). The shared kingship and the acute risk of the Czech Lands dividing ended with Matthias' death in 1490, when all the elements of the Czech Crown were again unified. Conflicts between Catholics and Calixtines ceased in 1485, when the Catholics accepted Calixtine demands. Although religious wars had left the Czech Lands totally devastated by the end of the 15th century, religious disputes diminished for some time (Macek, 2001).

Frequent wars, disease including plague epidemics, famines, and a falling birth-rate resulted in a marked depopulation of the Czech Lands; the number of inhabitants fell by as much as 40%–50% (Čornej, 2003). This trend was especially evident in villages (many simply ceased to exist); it was less severe in small towns and cities. However, by the end of the 15th century the demographic situation showed signs of improvement (Fialová, 1996).

**3 Data**

**3.1 Documentary data**

Weather-related information concerning the Czech Lands during the 15th century may be found in the following main documentary sources:

**(i) The Old Czech Annals**





The Old Czech Annals (*Staré letopisy české*), collated by František Palacký in 1829, consist of 33 manuscripts, critically

analysed by Čornej (1988) and republished by Černá et al. (2003, 2018). Closest to the archetype of these annals are the

Latin records of an anonymous citizen of the Old Town of Prague (see Fig. A1 for location of places), who presented events

for 1419–1432 CE. It was translated into Czech and later extended to 1440. The Annals were continued by further records

from the citizens of Prague then, in the last quarter of the 15th century, by east-Bohemian chronicles originating in Hradec

Králové. The Old Czech Annals end in 1527 CE. Much of the material contained within them is devoted to everyday events

in the above towns (and beyond) and, naturally to the weather, together with related phenomena, the quality and quantity of

harvests, fluctuations in prices, and more. For example, the winter of 1434/1435 is described as (Černá et al., 2003, p. 97):

"*The winter was cruel this year and large quantities of snow* [fell]*, more than could be remembered for the past hundred*

*years. Due to the snow, people were unable to walk or travel from town to town and from villages to towns. And that winter*

*started on Saint Andrew's day* [30 November 1434] *and continued as far as to the end of February* [1435]. *The thaw was not*

*sudden, so there was no "huge water", despite the huge quantity of snow*."

**(ii) chronicles**

The records kept by Bartošek of Drahonice (Goll, 1893), written in Latin by a retired soldier living in Karlštejn Castle in

central Bohemia, provide an example of 15th-century chronicles. They cover the years 1419–1443 CE. His description of a

great windstorm in 1435 is typical (Goll, 1893, p. 618): "*In this year* [1435]*, on the Thursday* [14 July] *before the days of the*

[celebration of the] *Dispersion of the Apostles, around the hour of vespers, a great windstorm of enormous force, lasting half*

*an hour, appeared, such as had not occurred for 30 years, and did much damage to castles and houses, carried away roofs*

*and demolished various buildings. In some regions of Bohemia and particularly in Knín, hailstones bruised cereals and did*

*great damage*."

**(iii) letters**

Some weather-related reports may be found in various written communications. For example, Enea Silvio Bartolomeo

Piccolomini, later Pope Pius II, the imperial emissary, included a remark about the cold and rainy weather in Bohemia in a

letter dated 30 August 1451, written during his journey to an assembly (Wolkan, 1912). Other letters report, for example,

damage done by floods, or economic difficulties arising out of droughts.

**(iv) accountancy books**

Financial records of accounts contain information about municipality expenses. *Liber rationum* from Louny, preserved in

fragments for 26 October 1450–3 February 1472 (missing data particularly between 23 July 1464 and 6 February 1469) and

22 November 1490–5 September 1491 (Vaniš, 1979; Brázdil and Kotyza, 2000), features the regular Saturday payments of

wages for work done in the preceding week. The nature of some of this work (for example, the beginning and the course of

haymaking, grain harvest and grape harvest, clearing ice from the River Ohře or watermills) depended on the character of the

preceding weather. *Liber rationum* also indicated any lack of income from watermills when the Ohře flooded, as noted for 7

March 1457 (Vaniš, 1979, p. 486): "*Nothing* [no income] *from mills due to flood*."

**(v) non-contemporary sources**



Records originating later than the events to which they refer, often based on unknown previous sources, may be used after critical evaluation, provided they do not contradict other sources and/or are confirmed by other primary sources from the

Czech Lands or neighbouring countries. The chronicle kept by Václav Hájek of Libočany from 1541 (Hájek, 1541), for which weather data were critically evaluated by Brázdil and Kotyza (1995), and a calendar by Daniel Adam of Veleslavín published in 1590 (archival source AS1), serve as examples.

### 3.2 Climatic data

In order to compare Czech 15th-century climatic data derived from documentary sources with others, the following

reconstructions based on tree-ring (TRW) data collected directly over the Czech Lands territory were used:

(i) March–July precipitation totals reconstructed from fir (*Abies alba*) TRWs for South Moravia (Brázdil et al., 2002);

(ii) May–July precipitation totals reconstructed from oak (*Quercus spp.*) TRWs for Bohemia (Dobrovolný et al., 2018);

(iii) June–August self-calibrating Palmer Drought Severity Index (scPDSI) reconstructed from oak (*Quercus spp.*) $\delta^{18}$O and $\delta^{13}$C stable isotopes (Büntgen et al., 2021);

(iv) June–August water balance reconstructed from the $\delta^{13}$C data used in (iii) and annual temperature sum (sum of temperatures for days > 10 °C) from $\delta^{13}$C data used in (iii) (Torbenson et al., 2023).

To characterize the Central European context of the 15th-century climate, two climate reconstructions were used:

(i) European June–August temperature multi-proxy reconstruction by Luterbacher et al. (2016) covering the last two millennia;

(ii) gridded monthly paleo-reanalysis ModE-RA (Valler et al., 2024) covering the 1421–2008 CE period and characterizing globally spatiotemporal variability of temperature, precipitation, and sea level pressure.

To describe external forcings and important circulation modes of the 15th century, the following series were used:

(i) total solar irradiance (TSI) by Lean (2018);

(ii) mean Northern Hemisphere stratospheric volcanic aerosol optical depth (AOD) by Toohey and Sigl (2017);

(iii) tree-ring reconstruction of the winter (December–March) reconstruction of the North Atlantic Oscillation (NAO) by Cook et al. (2019).

### 4 Methods

The primary documentary sources reported in Sect. 3.1 were analysed to describe the course of the weather and its anomalies in the Czech Lands during the 15th century. Reports relating to the weather from secondary sources were used only when

they coincided with information from primary sources, and/or were supported by data from neighbouring countries such as Germany, Poland and Austria. With respect to comparability with recent climate, the data presented were recalculated from the Julian calendar to the current Gregorian style by adding nine days to the original dates. Quotations of data sources are based on published versions of the relevant documents, except for those that were available only as archival documents.

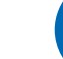

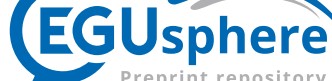

The documentary data concerning the weather and related phenomena were used to interpret past weather patterns and
provide their expression in the form of temperature and precipitation indices (Pfister, 1992; Brázdil et al., 2005). Since the
data quality was inadequate for greater detail, a simple classification of months on a three-degree scale was employed: –1 –
cold, 0 – normal and +1 – warm for temperatures, and –1 – dry, 0 – normal and +1 – wet for precipitation. A seven-degree
scale was used to interpret seasonal indices (winter – DJF, spring – MAM, summer – JJA, autumn – SON): temperature: –3
– extremely cold, –2 – very cold, –1 – cold, 0 – normal, +1 – warm, +2 – very warm, +3 – extremely warm (mild in winter);
precipitation: –3 – extremely dry, –2 – very dry, –1 – dry, 0 – normal, +1 – wet, +2 – very wet, +3 – extremely wet.

The series of proxy reconstructions and external forcings used for comparison with documentary data from the Czech Lands
during the 15th century were generally adjusted to anomalies from the reference period 1961–1990 and supplemented with a
low-pass Gaussian filter. The 1961–1990 reference was prioritised over the more recent period 1991–2020, which is much
more strongly influenced by recent warming (Brázdil et al., 2022b). The significance of differences in reconstructed
temperature and precipitation between the 15th century and 1961–1990 was tested by t-test (differences in means) and
Levene's test for differences in variance (Levene, 1960). Additionally, the proxy reconstructions of hydroclimatic variability
for years identified as having summer precipitation deficits (i.e., –1, –2, or –3) in the documentary data were fitted with
Kernel density distributions and compared to the same distributions of reconstructed values for other 15th-century years. A
two-sample Kolmogorov–Smirnov test (Simard and L'Ecuyer, 2011) was applied to assess whether the estimates from the
two groups are from different continuous distributions.

Using the ClimeApp application (Warren et al., 2024), we calculated the standard deviation (SD) ratio, which helps clarify
the differences found from the above tests using ModE-RA reanalysis. The SD ratio is defined as the standard deviation of
the ModE-Sim model ensemble divided by the standard deviation of ModE-RA after the assimilation of observations, and it
shows to what extent the model is constrained by observations. The smaller the SD ratio is than 1 (no constraint), the more
weight the proxy data and observations have in the final assimilated values of the variable under study.

Composite analysis (von Storch and Zwiers, 1999) was used to describe typical climatic patterns, including estimates of the
prevailing atmospheric circulation during (i) cold winters, (ii) warm-dry summers and (iii) cold-wet summers in Central
Europe during the 15th century. Based on the Czech indices, we selected these three types of extreme seasons and compiled
the composite maps of mean sea level pressure, air temperature, and precipitation totals in the form of anomalies from the
1961–1990 reference. Because of the small number of Czech seasons in the analyzed composites, we evaluated significant
differences of the composite maps from the reference using the nonparametric bootstrap test (Efron and Tibshirani, 1994).
The test compared 1000 randomly simulated samples with replacement for each type of the extreme season.

Finally, Spearman's rank correlation and regression analysis were used to quantify the common variance between the Czech
temperature and precipitation indices and the reanalysis in 1422–1499 CE. These analyses were limited only to the years
with available indices interpreted from documentary data.



# 5 Results

## 5.1 Temperature and precipitation indices

Weather patterns and related phenomena in the Czech Lands for individual years within the 15th century, extracted particularly from narrative documentary evidence, are described in detail in the Supplement. They can be complemented by

*Liber rationum* from the town of Louny (Sect. 3.1, point iv), which is a particularly important source of phenological data for certain years of the 15th century. As demonstrated by Vaniš (1982) and Brázdil and Kotyza (2000), its records may be used for weather interpretation based on the dates for routine annual agricultural tasks; these, of course, were dependent on the weather patterns of the preceding days. Data from 1451–1463 and 1469–1471 CE enabled calculation of the mean dates upon which field work started and ended (Table 1); their inter-annual fluctuations appear in Fig. 1. Particular attention was

paid to field work advanced or delayed by more than a week from the mean date, which may reflect weather patterns before or during haymaking, grain and grape harvests, reflecting deviations from the usual annual agricultural cycle (cf. Brázdil et al., 2019). As is evident from the values for standard deviation in Table 1, the start of grain harvest was the most stable, but its ending proved the most variable.

**Table 1. Dates upon which wages were paid in the Louny region for field work in the preceding week during 1451–1463 and 1469–1471 CE according to *Liber rationum*. Key: B – beginning, E – end, n – number of years with related records, MD – mean date, STD – standard deviation (days), ED – earliest day, LD – latest day (corrected according to Brázdil and Kotyza, 2000).**

| Work | n | MD | STD | ED | LD |
|---|---|---|---|---|---|
| Haymaking – B | 15 | 7 July | 12.8 | 9 June 1455 | 30 July 1470 |
| Haymaking – E | 11 | 7 August | 11.5 | 21 July 1455 | 21 August 1452 |
| Grain harvest – B | 12 | 14 July | 7.2 | 28 June 1456 | 23 July 1453, 1459 |
| Grain harvest – E | 12 | 9 September | 22.1 | 15 August 1457 | 8 November 1456 |
| Grape harvest | 15 | 20 October | 10.9 | 30 September 1471 | 8 November 1451 |





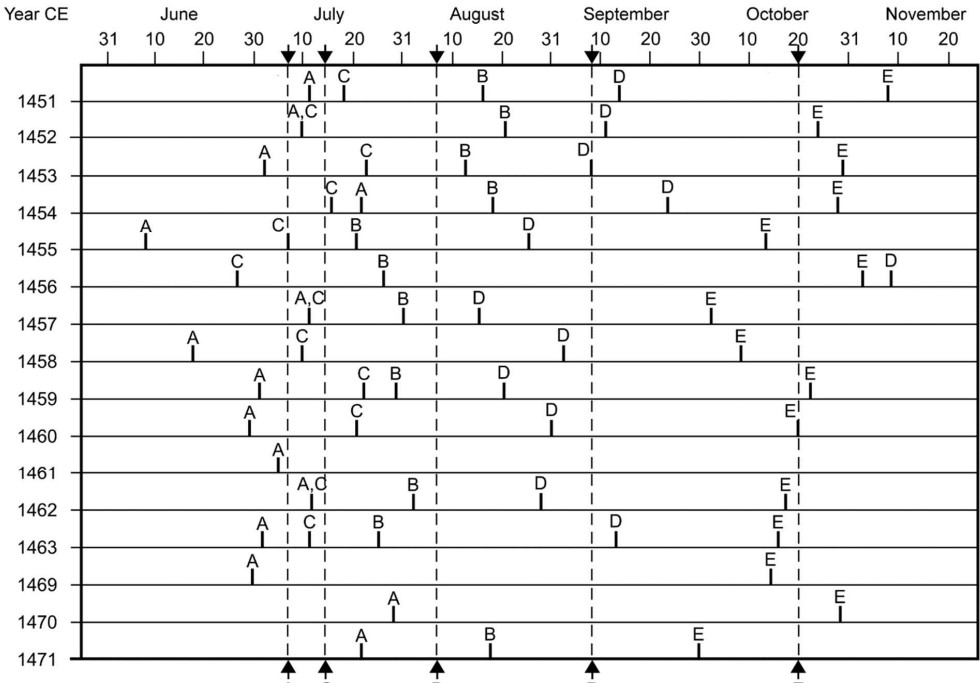

**Figure 1. Changes in the beginnings and endings of haymaking, grain harvest and grape harvest (always in the week before the given date) in the Louny region for 1451–1463 and 1469–1471 CE according to *Liber rationum*: A – beginning of haymaking, B – end of haymaking, C – beginning of grain harvest, D – end of grain harvest, E – beginning of grape harvest. Vertical broken lines indicate mean dates in 1451–1471 (Table 1) (corrected according to Brázdil and Kotyza, 2000).**

Weather data from the described sources (see Supplement) were used for quantification of monthly and seasonal temperature and precipitation indices. Monthly indices were interpreted on a 3-degree scale and seasonal indices on a 7-degree scale (see Sect. 4). Since documentary evidence for the Czech Lands during the 15th century is relatively scarce, the months and seasons interpreted represent only a small part of that time. However, at least one month and/or season was interpreted in 45 years for temperature indices and in 40 years for precipitation indices (Fig. 2). Weather patterns for winters and summers were more frequently accessible to analysis than those for springs and autumns. This is predominantly a response to the practicalities of a largely rural society that needed to use its roads for essential business and diplomacy. Thus, winter attracted attention and comment if the weather was extreme – severe frosts or a huge amount of snow causing fatalities, blocking transport and putting watermills out of commission, or if it was very mild and muddy, with no snow or very little. In summer, direct effects on agricultural activities were paramount, with possible harvest failure, together with consequences such as increasing prices, the occurrence of famine, etc. Spring tended to manifest as either prolongation of a severe winter

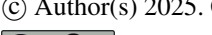


to March or April on the one hand, or the effects of May weather on agricultural work, on the other. Autumn reports tended

to mainly refer to the weather's influencing the start of the grape harvest or an advanced onset of winter frosts.

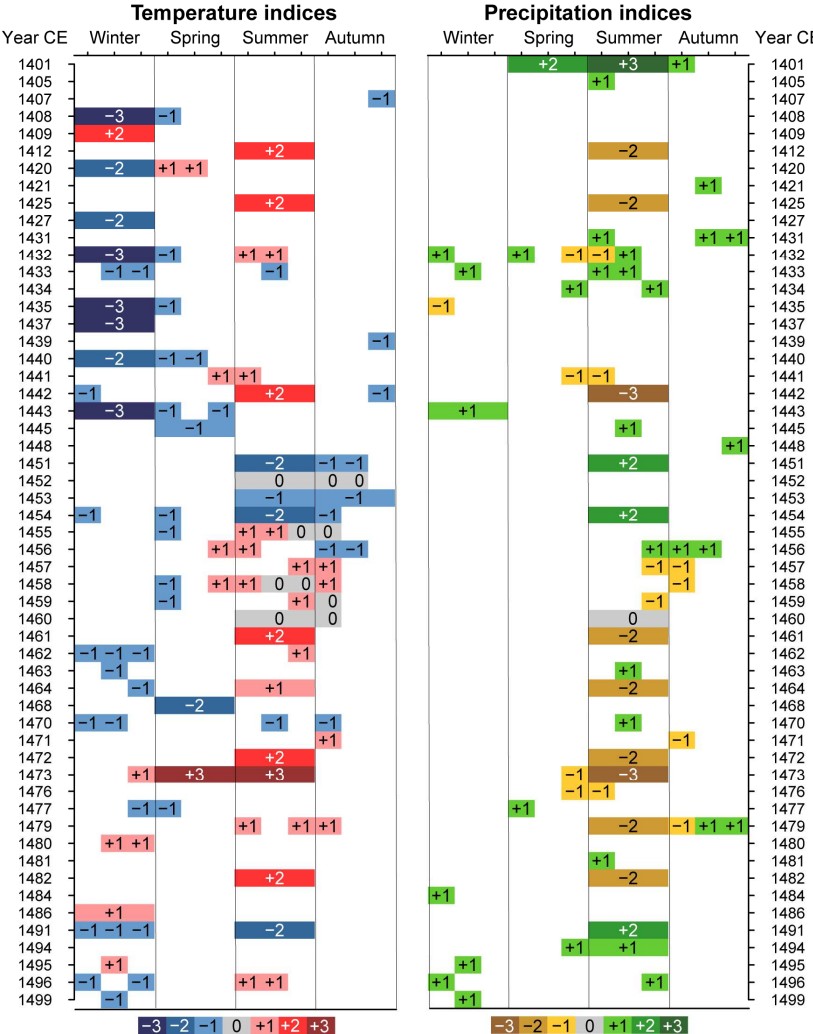

**Figure 2. Monthly and seasonal temperature and precipitation indices reconstructed from Czech documentary evidence for the 15th century.**





**5.2 Hydrometeorological extremes**

Czech documentary sources also contain frequent records of HMEs and their impacts. This holds particularly true of the 25 reported floods that occurred during the course of 18 years. The majority of them could be classified as March/April floods after severe winters, the result of snowmelt caused by sudden warming and rain, sometimes combined with ice-floes jamming in riverine structures and blocking the flow. Rainy floods resulting from very heavy rain on one or several days

tended to appear in summer. Several floods, together with their local effects, can be identified as flash floods resulting from torrential rain, causing damage at local and even regional levels. Of particular note is the year of 1432 CE, with three floods, in March, July and December. A winter flood, a flash flood and a flood caused by three days of continuous rain were recorded in 1496. Two floods in one year also occurred in 1405, 1433, and 1445. The records kept in *Liber rationum* indicate five winter floods on the Ohře at Louny: 1457, 1458, 1459, 1461 and 1464. A notable period, with seven floods, was

identified between 1431 and 1434, followed by a further flood in 1436.

The most remarkable flood occurred at the end of July 1432 in Bohemia. The flood followed heavy rains on 31 July, interrupting the dry period that had persisted from 2 May to 28 July (Goll, 1893). The July patterns in the Atlantic–European region were generally characterised by a weakening of the Azores High and a pressure decrease extending over Central Europe, while a pressure increase occurred only in the northern part of the region (Fig. 3a). One area of increased

precipitation extended over Central Europe, while two others covered the British Isles and the area adjacent to the eastern Baltic Sea (Fig. 3b). The River Vltava flooded the Old Town of Prague and low-lying parts of the Lesser Town (*Malá Strana*), and the inhabitants could only move around in boats. The water swept away all the mills on the Vltava and destroyed houses. Many people drowned and much livestock was lost. Timber carried downstream by the river built up against Charles Bridge, which was made of stone, and brought down five arches. Damage to the stone bridge on the Otava at

Písek and two bridges on the Vltava at Český Krumlov was also reported. The remnants of houses, mills, furniture, and other debris were deposited on the plains around the river. Apart from the Vltava at Prague, the Berounka at Beroun and areas around the Elbe were also flooded. Cereals in the fields were flooded or swept away. Such a flood was beyond the memories of even old men "[since the time of] *the flood of the world*" (Höfler, 1856; Goll, 1893; Horčička, 1899; Šimek, 1937; Palacký et al., 1941; Šimek and Kaňák, 1959). On the same day, flooding ravaged Moravia, Austria and Hungary (Höfler,

1865b). The July 1432 flood was considered a possible "millennial flood", comparable with the most recent devastating flood of August 2002 in Bohemia (Brázdil et al., 2006).



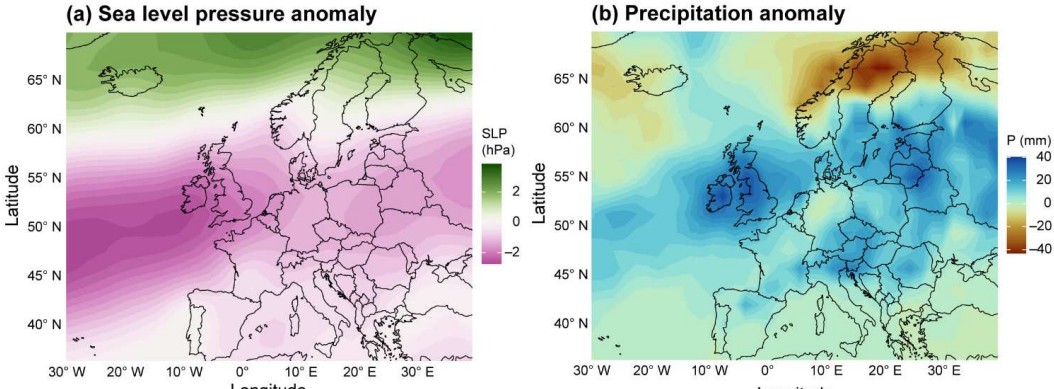

**Figure 3.** Anomalies (w.r.t. 1961–1990) in (a) sea level pressure SLP (hPa) and (b) precipitation totals P (mm) in the Atlantic–European region in July 1432 according to ModE-RA reanalysis (Valler et al., 2024).

Other damaging weather events described in Czech documentary sources may be attributed to convective storms, possibly accompanied by torrential rain causing flash floods, by hailstorms, severe winds, and lightning strikes. For example, Prague suffered a notable thunderstorm in the night 1/2 July 1499 and lightning struck the monastery of the Holy Spirit and set its roof on fire. According to the annalist "*there was very great and frequent thunder and lightning that whole night, almost incessant*" (Palacký et al., 1941, p. 217). Excluding the four flash floods, convective storms were described in a further 21

years, reported especially in the 1490s (six years) and in the 1470s (five years). For example, on 8 July 1474, a storm in Cheb damaged houses, barns, and trees in gardens in the town and forests in its surroundings (Gradl, 1884). On the same day, lightning during a thunderstorm in Jihlava started a fire in which 67 of the most important houses in the town were destroyed and 20 people burned to death (d'Elvert, 1861). A storm was also recorded in Moosbach, Bavaria (Klemm, 1983) and in Melk in Lower Austria (Wattenbach, 1851).

Some strong winds of greater territorial extent, attributed to windstorms arising out of large horizontal pressure gradients, were reported only for 1412, 1432 (two cases), 1441 and 1495. For example, on 4 December 1412, a terrible windstorm, beyond living experience at the time, destroyed many buildings as it particularly carried away roofs, and broke and uprooted trees in gardens and woods all over Bohemia (Šimek, 1937; Palacký et al., 1941; Dušek, 1993; Černá et al., 2003).

**5.3 Czech temperature and precipitation indices in a broader context**

Incomplete precipitation indices derived from Czech documentary data can be partly complemented by dendroclimatological analysis of fir and oak tree-ring series. The wettest period in the presented indices coincides with the relatively cold climate of the 1430s, as derived from documentary evidence for Central and Western Europe (Camenisch et al., 2016), as well as with lower values of temperature sums derived from oak δ18O (Fig. 4a). Four proxy-based drought and precipitation reconstructions for the 15th-century Czech Lands (Fig. 4b-e) exhibit both similar features and differences to the derived



precipitation anomalies, at inter-annual and inter-decadal scales. The two precipitation reconstructions agree on the driest decade centered on the 1420s (Fig. 4b,c). The 1470s drought is also well-expressed in South Moravian precipitation series, revealed as the second driest decade in the 15th century (Fig. 4b). The wettest decades varied in time and space: in 1450–1459 in the March–July precipitation totals for South Moravia and in 1405–1414 in the May–July precipitation for Bohemia (Fig. 4b,c). Reconstructed series show some coherent signals, especially between the 1430s and 1460s, when their running

correlations are significant and reach 0.3–0.5 (not shown here). However, the common signal is much weaker before and after that period. This may be attributed to the fact that all these reconstructions refer to different spatial domains and slightly different seasons.

    In the scPDSI reconstruction, a well-expressed, long-term drying tendency from the wettest decade of 1428–1437 to the end of the 15th century is apparent (Fig. 4d). In contrast, the driest decade (1471–1480) includes a warm beginning to the 1470s

with 1473 characterised as particularly hot and dry (Camenisch et al., 2020). Moreover, the last decades of the 15th century (together with the beginning of the 16th century) are among the most severe drought periods in the Common Era. The intensity of this drought spell is comparable with the most recent extremely dry period, in the second part of the 2010s (Büntgen et al., 2021).



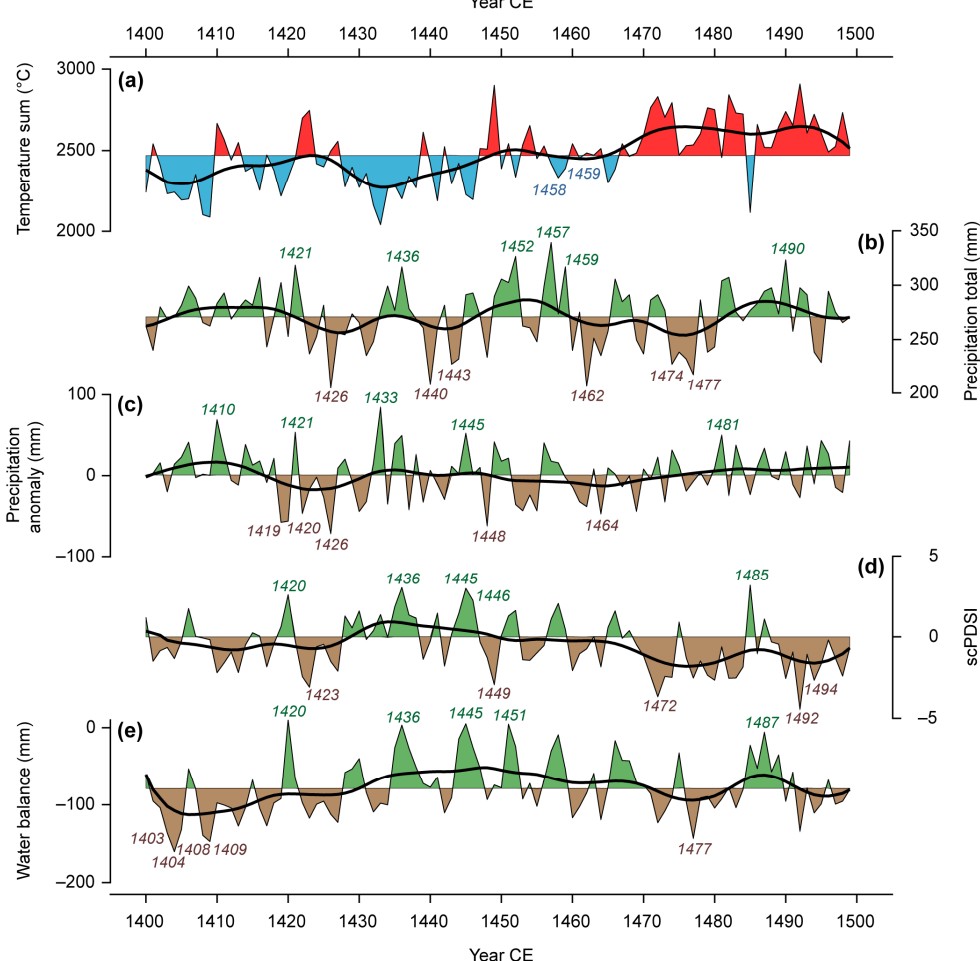

**Figure 4. Estimated variability of climate variables over the territory of the Czech Lands in the 1400–1499 CE period: (a) annual temperature sum from oak (*Quercus spp.*) δ¹⁸O stable isotopes (Torbenson et al., 2023); (b) March–July precipitation totals reconstructed from fir (*Abies alba*) TRWs for South Moravia (Brázdil et al., 2002); (c) May–July precipitation totals reconstructed from oak TRWs for Bohemia (Dobrovolný et al., 2018); (d) June–August self-calibrating Palmer Drought Severity Index (scPDSI) reconstructed from oak δ¹⁸O and δ¹³C (Büntgen et al., 2021); and (e) June–August water balance from oak δ¹³C (Torbenson et al., 2023). Anomalies of mean annual values with respect to the 15th-century mean (with the exception of scPDSI, which is plotted in absolute values) are expressed in green and brown colours (in red and blue for temperature sums). Bold black lines indicate series smoothed by a low-pass Gaussian filter for 15 years. For each of the hydroclimate series (b)–(e), the five wettest and five driest years are indicated.**



The two precipitation reconstructions based on tree-ring width (TRW) do not show similar long-term tendencies but rather a
predominance of high inter-annual variability (Fig. 4b,c). The lack of low-frequency agreement may be related to the fact
that TRWs as a proxy-type generally suppress long-term fluctuations compared with chronologies compiled from stable
isotopes (Büntgen et al., 2020). Moreover, the scPDSI, as presented in Fig. 4d, is considered a reliable indicator of long-term
drought (Brázdil et al., 2016). Conversely, the scPDSI reconstruction by Büntgen et al. (2021) combines $\delta^{13}$C and $\delta^{18}$O
series, variables that have been argued to reflect different signals (Torbenson et al., 2023). The separate reconstructions of
water balance and growing season temperature suggest that the positive scPDSI values in the 1430s were mainly driven by
low temperatures (and the subsequent weakening of evapotranspiration demands) rather than high precipitation – which
aligns with the temperature information from Camenisch et al. (2016). However, it should be noted that the $\delta^{13}$C-only water
balance reconstruction also displays a drying trend from the late 1430s (Fig. 4e).

The degree of agreement between precipitation indices derived from documentary sources (Fig. 2) and four proxy-based
hydroclimate reconstructions (Fig. 4b-e) was evaluated for five of the driest (wettest) years identified in reconstructed series.
As follows from Table 2, full or only partial agreement appears for just a few of them. These varying relationships may
reflect the fact that precipitation extremes tended to be spatially restricted to the extent that they found no reflection in
documentary sources. Further, TRW-based precipitation reconstructions display relatively low skill. Finally, some extreme
years or seasons are not reflected in precipitation indices simply because of the low density of available documentary
evidence.

**Table 2. Overview of the five driest/wettest years in four hydroclimatic recontructions for the Czech Lands in the 1400–1499 period. Years of part or full agreement with Czech precipitation indices (Fig. 2) appear in bold type. Six years are provided for the Brázdil et al. (2002) reconstruction due to equal ranks of reconstructed values.**

| Reconstruction | Driest years | Wettest years |
|---|---|---|
| March–July precipitation (Brázdil et al., 2002) | 1426, 1440, 1443, 1462, 1474, 1477 | 1421, 1436, 1452, **1457**, 1459, 1490 |
| May–July precipitation (Dobrovolný et al., 2018) | 1419, 1420, 1426, 1448, **1464** | **1410**, 1421, 1433, **1445**, 1481 |
| Summer scPDSI (Büntgen et al., 2021) | 1423, 1449, **1473**, 1492, 1494 | 1420, 1436, **1445**, 1446, 1485 |
| June–August water balance (Torbenson et al., 2023) | 1403, 1404, 1408, 1409, 1477 | 1420, 1436, **1445**, **1451**, 1487 |


Despite only partial matches for the most extreme years in the proxy-based reconstructions (Table 2), the distributions of
some tree ring-based hydroclimatic estimates differ significantly between dry and wet years identified in the documentary
data (Fig. 5). All four records have lower means in the dry years compared to the wet years, although the difference is minor



for the March–July precipitation reconstruction (Brázdil et al., 2002; Fig. 5a). Both the May–July precipitation (Dobrovolný

et al., 2018; Fig. 5b) and the water balance (Torbenson et al., 2023; Fig. 5d) estimates display statistically significant

differences in their distributions for dry and wet years. Although the difference is smaller for scPDSI (Fig. 5c), this muted

disparity may be due to temperature – as the temperature sum estimates indicate a positive trend over the 15th century (Fig.

4a), and the predictor (i.e., $\delta^{18}O$ of *Quercus* tree rings) is included in the scPDSI reconstruction of Büntgen et al. (2021). The

relative influence of temperature in scPDSI (as compared to the water balance reconstruction) is especially evident in the

1480s and 1490s (Fig. 4a), for which the documentary data indicates several years of wetness (Fig. 2).

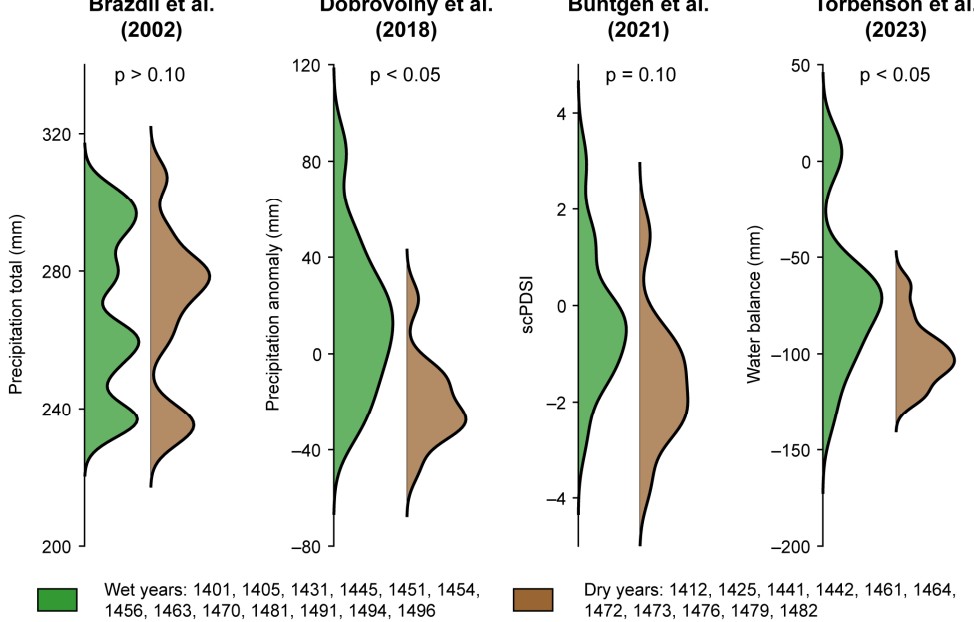

**Figure 5. Kernel density distributions (based on bandwidths = 1/10th of the range of reconstructed values) of proxy-reconstructed hydroclimate for years of summer precipitation anomalies in the documentary data (green = wet; brown = dry) of the 15th century. Significant values (p < 0.05) indicate that the values in Dobrovolný et al. (2018) and Torbenson et al. (2023) are from**
**different distributions.**

**5.4 The 15th century climate in Central Europe**

Depending on differing views on the beginning of the Little Ice Age (e.g., Miller et al., 2012; Wanner et al., 2022), the 15th

century can be understood either as part of it or as a transition period leading to its onset. In evaluating the climate of this

century, two external forcings – solar and volcanic activity – should be considered. A substantial part of the 15th century

falls within the Spörer Minimum of solar activity (Eddy, 1976). Although some authors define this solar minimum

differently in terms of its beginning and end (Jiang and Xu, 1986), low solar activity during this century is evident from Fig.





6a. As for volcanic activity, large volcanic eruptions were identified particularly in the 1450s (Fig. 6b). A reconstruction of the North Atlantic Oscillation (Cook et al., 2019), representing a considerable component of internal climate variability in Europe, reveals slightly lower values during the 15th century compared to the previous and following centuries (Fig. 6c).

The same long-term NAO variability with more pronounced period of lower values in the middle of the century confirms a model-tested NAO reconstruction by Ortega et al. (2015) (see Fig. A2).

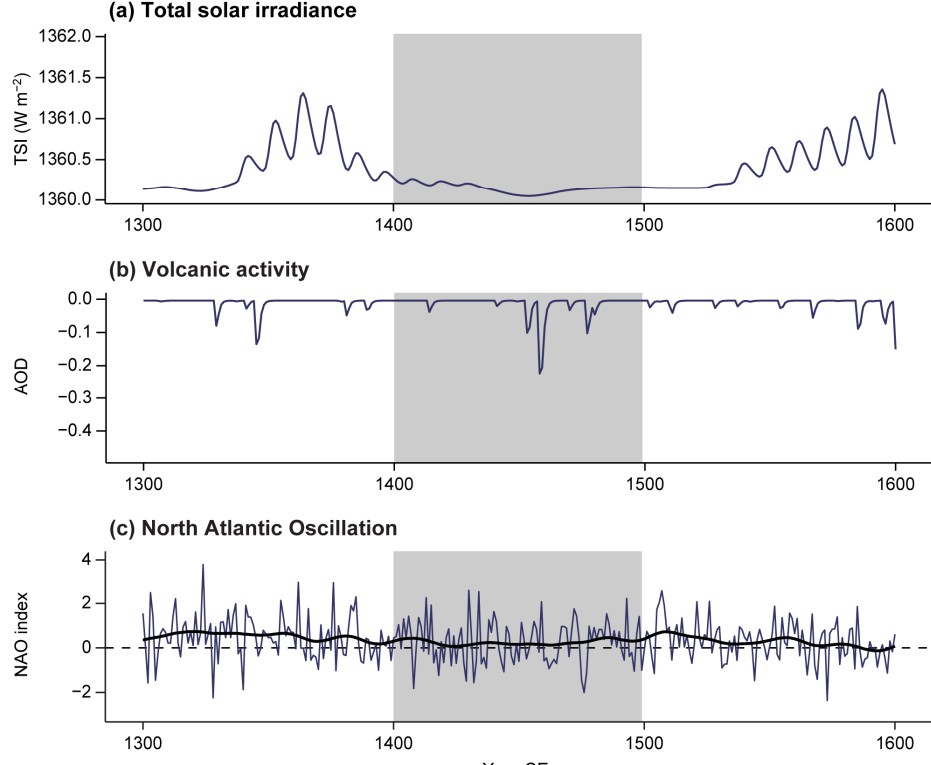

**Figure 6. Fluctuations in (a) total solar irradiance (TSI; Lean, 2018), (b) volcanic activity (stratospheric aerosol optical depth AOD; Toohey and Sigl, 2017), and (c) North Atlantic Oscillation (NAO index; Cook et al., 2019) in the 1300–1600 CE period. Note**
**inverted AOD values. NAO index is smoothed by a 20-year Gaussian filter.**

Central European climate of the 15th century at the seasonal and annual levels can be described using the ModE-RA reanalysis by Valler et al. (2024). Although this reanalysis does not cover the first two decades (starting from 1422 CE), its spatial resolution allows characterizing temperature and precipitation variability in Central Europe for all seasons (Fig. 7a,b). The temperature variability since the 1420s shows significantly lower temperatures in all seasons except JJA compared to

the reference period 1961–1990 (Fig. 7c). Precipitation totals were significantly higher compared to the reference in all



seasons, especially in JJA and SON (Fig. 7d). Variances of 15th-century temperatures and precipitation were significantly lower compared to 1961–1990, with the only exception of JJA temperatures. The outliers identified in the reanalysis data on the Central European scale and compared with the Czech extremity indices (Fig. 2) confirm only the exceptionality of the year 1473 for MAM. Other outlier years do not appear in the series of Czech temperature and precipitation indices, which

may be mainly related to the low density of records in most seasons.

The variability of seasonal temperatures and precipitation totals in the period 1422–1499 was significantly smaller (p < 0.01) than in the period 1961–1990 in all seasons except JJA in the case of temperature (not shown). The low variability of temperature and precipitation compared to the reference period may be related to the fact that the period 1422–1499 is at the beginning of the ModE-RA reanalysis. In this initial period, the resulting values of temperature and especially precipitation

are largely generated by the model used and are less constrained by the assimilated observations. This is shown by the SD ratio, which for the Central European region is less than 0.5 for temperature and less than 0.8 for precipitation.





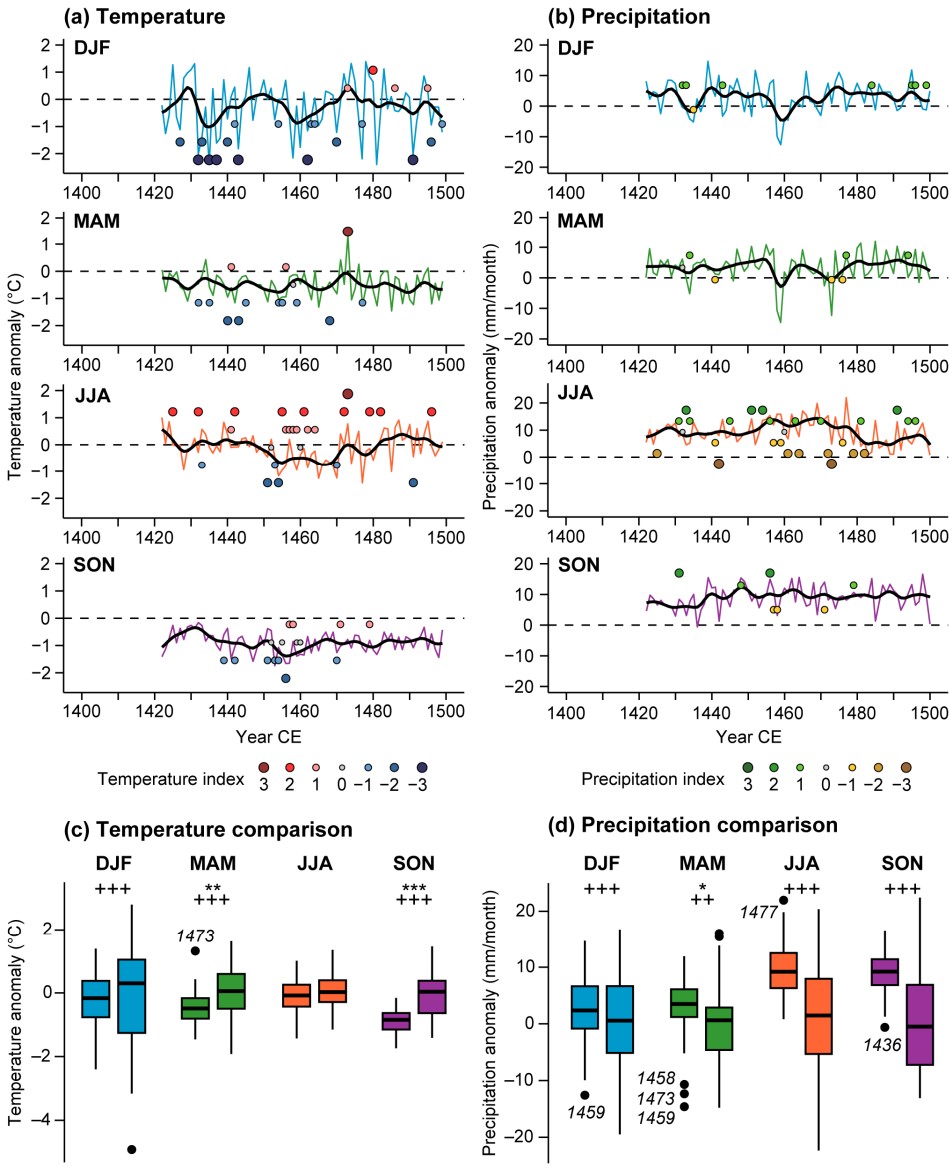

**Figure 7. Fluctuations in mean seasonal temperatures (a) and precipitation totals (b) in Central Europe (45–55° N, 5–25° E) in the 1422–1499 CE period according to ModE-RA reanalysis (Valler et al., 2024) complemented with the Czech seasonal temperature and precipitation indices. Values are expressed as anomalies w.r.t. 1961–1990 (dashed horizontal line) and smoothed by a 10-year**



**Gaussian filter. Comparison of mean seasonal temperatures (c) and precipitation totals (d) in the 1422–1499 CE period (left boxplot) with the 1961–1990 reference (right boxplot), and completed with t-test comparing means (\*\*\*: p < 0.001; \*\*: p < 0.01; \*: p < 0.05) and Levene's test comparing variances (+++: p < 0.001; ++: p < 0.01; +: p < 0.05). Black points indicate temperature and precipitation outliers.**

The millennial context of 15th-century climate in Central Europe can be derived from different JJA reconstructions based on natural proxies. This includes, for example, the reconstructed JJA temperatures by Luterbacher et al. (2016) in Fig. 8a. While during the first half of this century JJA temperatures were comparable to those in 1961–1990, the second half of the 15th century was characterised by a significant increase from the minimum in its middle. This may be considered a response to the synergic development of the main forcings towards cooler climatic patterns. The Czech scPDSI reconstruction by

Büntgen et al. (2021), which captures multi-decadal variability well (often missing in TRW-based reconstructions), shows a clear decline during the 15th century, leading to a well-expressed dry period centred at the beginning of the 16th century (Fig. 8b).

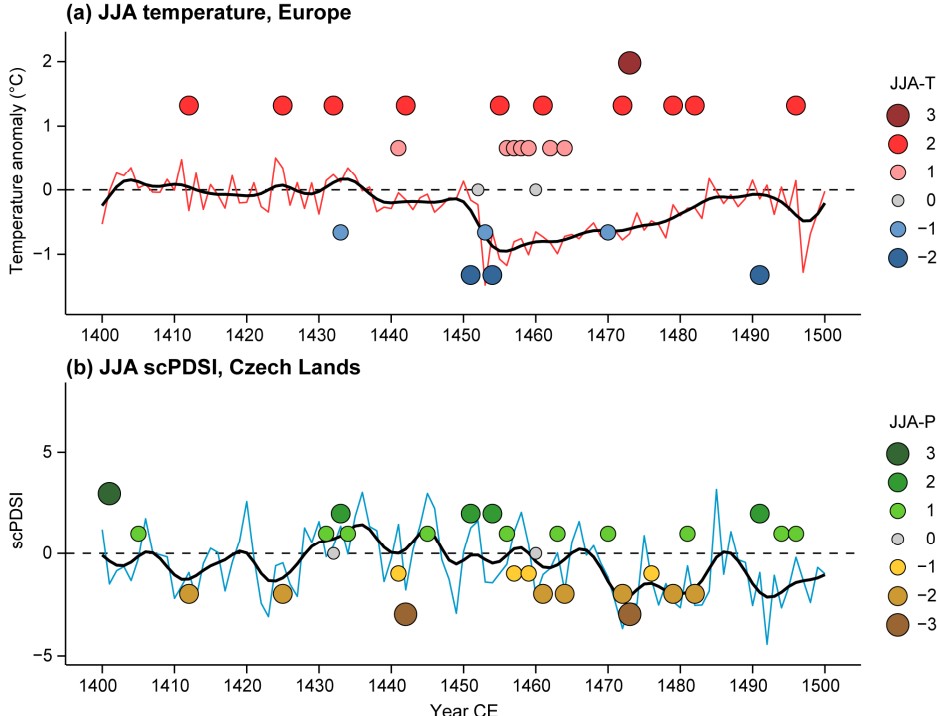

**Figure 8. Fluctuations in (a) European JJA temperatures (Luterbacher et al., 2016) and (b) Czech JJA scPDSI (Büntgen et al., 2021) in the 1400–1499 CE period complemented with the Czech summer temperature (JJA-T) and precipitation (JJA-P) indices. Temperatures are expressed as anomalies w.r.t. the 1961–1990 reference (dashed horizontal line). Both series are smoothed by a 10-year Gaussian filter.**



### 5.5 Linking extremes to atmospheric circulation

The occurrence of particularly cold winters and warm-dry summers in Central Europe indicates a predominantly continental
climate, which manifests itself in typical patterns of atmospheric circulation. Relatively higher frequency of winter and
summer indices as well as higher number of seasons whose temperature and precipitation conditions differed significantly
from the "normal" conditions (seasons with the index values higher than 1 and lower than –1) made it possible to use
composite analysis to determine whether the occurrence of such exceptional seasons in the Czech Lands during the 15th
century was also characterized by typical circulation conditions. Existing 15th century Czech indices summarized in Fig. 2
allowed compilation three types of composites: cold winters, warm-dry summers, and cold-wet summers. Only seasons with
the temperature (precipitation) index of grades 2, 3 (or –2, –3 respectively) were included to the composite. Then we used
ModE-RA data and compiled mean fields of sea level pressure, temperature and precipitation from the selected seasons (see
Section 4 for details). The mean composites were expressed as anomalies with respect to 1961–1990 and significant
deviations from the reference were tested using bootstrap test (Fig. 9). Additionally, we compiled composites from NAO
reconstruction to reveal underlying patterns of cold winters (Fig. A2). Finally, 15th century warm-dry summers and cold-wet
summers were projected to scPDSI JJA reconstruction (Büntgen et al., 2021; Fig. A3).



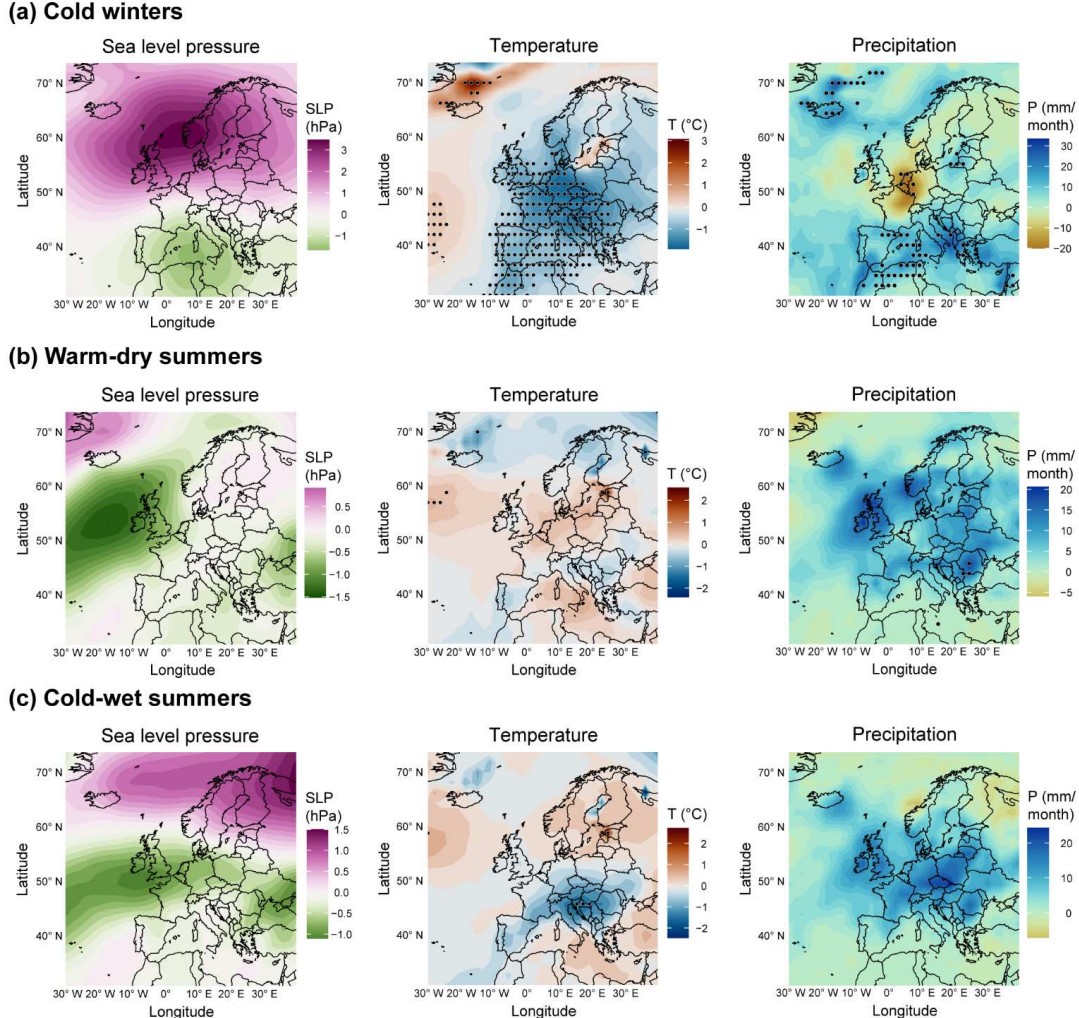

**Figure 9. Composite analysis of sea level pressure SLP (left column), temperature T (middle column) and precipitation P (right column) fields compiled from ModE-RA for (a) cold winters, (b) warm-dry summers and (c) cold-wet summers. Maps represent mean climate conditions in extreme seasons selected from the Czech documentary indices in the 15th century (cold winters: 1408, 1420, 1427, 1432, 1435, 1437, 1440, 1443, 1462, 1470, 1496; warm-dry summers: 1425, 1442, 1461, 1472, 1473, 1482; cold-wet summers: 1451, 1454, 1491) and are expressed as anomalies w.r.t. 1961–1990 reference. Black dots indicate grid-points with significant anomalies at a 95% significance level.**




The results of the composite analysis show the most consistent climatic conditions for cold winters. Their mean temperature
conditions were characterized by a clearly defined area with significantly below-normal air temperatures in Central Europe
and with significantly higher precipitation in Western Mediterranean (Fig. 9a, middle and right columns). The mean sea level
pressure field shows a typical dipole pattern with positive anomalies over the North Sea and southern Scandinavia and,
conversely, negative anomalies over the western Mediterranean (Fig. 9a, left column). This pattern is typical for the negative
phase of the NAO and for incursions of cold Arctic air from the northeast into Central Europe. To verify this, we also
applied the composite analysis to two reconstructed NAO series (Fig. A2). This figure clearly shows that cold winters
according to the Czech documentary indices well correspond with years of negative NAO values.

Warm-dry summers derived from the Czech documentary indices are also characterized by positive temperature anomalies in
Central Europe (Fig. 9b, middle column). The mean sea level pressure field is characterized by well-expressed negative
anomaly west of the British Isles, which supports warm airflow from the southwest to Central Europe (Fig. 9b, left column).
However, the mean precipitation distribution during warm-dry summers shows rather ambiguous results for Central Europe,
which may be related to the low coverage of precipitation data used in the reanalysis (Valler et al., 2024). Composite maps
for warm-dry summers show no significant deviations from the reference at the 95% significance level.

In contrast, the distribution of mean air temperature and precipitation, as well as the character of mean sea level pressure
field in Europe during Czech cold-wet summers, are quite consistent (Fig. 9c). The significant negative air temperature
anomaly in Central Europe is accompanied by above-mean precipitation. These conditions correspond well with the main
features of the pressure field distribution with well-expressed belt of lower air pressure centred on the latitude of 50°N.
These conditions are usually favourable for intense zonal flow and higher cyclonic activity in Central Europe.

Despite less frequent evidence of extreme summers in the Czech Lands in the 15th century, our analysis proved that the
occurrence of warm-dry summers according to the documentary sources clearly corresponded to summers with persistently
low scPDSI values (Fig. A3). Moreover, the mean scPDSI for warm-dry summers is significantly lower than the mean for
the cold-wet summers.

## 6 Discussion

Many of the weather events and anomalies, as well as the derived temperature/precipitation indices, reported herein for the
Czech Lands were also documented in other European regions or countries, such as Germany (Glaser, 2008), the Burgundian
Low Countries (Camenisch 2015) and the western-central European area (Pfister and Wanner, 2021). Interpretation of a
number of severe winters, extending to March or April in the Czech Lands during the 1430s (Fig. 2), together with the
occurrence of floods (also, in part, windstorms and convective storms), confirm the severe character of the cold 1430s in
Europe, as described by Camenisch et al. (2016). Pfister et al. (2024), analysing wine must quality as a reflection of weather
patterns for Germany, Luxembourg, eastern France, and the Swiss Plateau in 1420–2019 CE, identified the years 1470–1479
as having the highest average quality on the decadal scale, and 1453–1466 and 1485–1494 as years of poor quality,





attributed to prevailingly cold and wet summers. In the subsequent paper dealing with wine must yields for 1416–1988 CE, Pfister et al. (2025) identified as "good harvest" years those between 1416 and 1425 and further 1471–1473. Of the years of drought and low water levels in medieval Hungary (Kiss and Nikolić, 2015; Kiss, 2017), dry patterns in the Czech Lands tallied with those that they highlight in 1473, 1479 and 1482. The year of 1455, with a warm summer, was probably dry

there too.

The possibilities for direct validation of the Czech documentary indices and their use for reconstructing the climate of Central Europe in the 15th century are limited for a number of reasons. In the case of documentary sources from neighboring regions, direct comparison is usually limited to significant hydrometeorological extremes (Kiss, 2017; Camenisch et al., 2020). In the case of numerous proxy reconstructions based on natural archives, direct comparison is severely limited by

their seasonality, which is often restricted to late MAM and JJA (see Fig. 8). These are the main reasons why we used data from ModE-RA re-analysis (Valler et al., 2024) for comparison with the Czech seasonal documentary indices. Although this global reanalysis does not cover the entire 15th century, its benefit is the monthly resolution that allows direct comparison of the Czech indices from all seasons (see Fig. 7c,d). Furthermore, this is completely independent source in this study, as no data from the Czech Lands prior to 1500 CE has been assimilated in ModE-RA dataset. One disadvantage is that the density

of different types of proxies is relatively low in the 15th century and the re-analysis is dominated by the ensemble mean of the atmospheric circulation model in this period (see Hand et al., 2023 and Valler et al., 2024 for more details).

Czech DJF and JJA temperature indices and JJA precipitation indices (see Fig. 2), which are more frequent compared to other seasons, can be compared with the ModE-RA reanalysis by Valler et al. (2024). Regression analysis for years containing both Czech indices and average temperature and/or precipitation from the reanalysis for Central Europe, i.e., for

21 years (DJF temperature), 25 years (JJA temperature), and 27 years (JJA precipitation), was used for comparison. As shown in Fig. 10, good agreement exists for DJF temperatures (Spearman rank correlation $r_s = 0.78$, $p < 0.01$), and weaker agreement for JJA temperatures ($r_s = 0.43$, $p < 0.05$). This can be explained by the higher spatial homogeneity of DJF temperatures, significantly affected by NAO patterns at the pan-European scale (cf. Trigo et al., 2002), compared to JJA patterns, which are characterised by larger spatial variability due to weaker expressed circulation and stronger regional

effects (e.g., cloudiness, sunshine duration). Regional differences are also strongly enhanced in JJA precipitation patterns, with a dominant portion of localised convective precipitation ($r_s = 0.19$, $p > 0.05$). The very strong correlation between the DJF temperature indices and the reanalysis temperatures also confirms that the DJF indices contain the strongest climate signal and best calibrate the measured temperatures (cf. Table 3 in Dobrovolný et al., 2010). Moreover, the Czech temperature indices show very good agreement with the Burgundian Low Countries indices (Camenish, 2015). For a total of

21 years for which indices from both countries are available in the 15th century, they agree in 16 years.



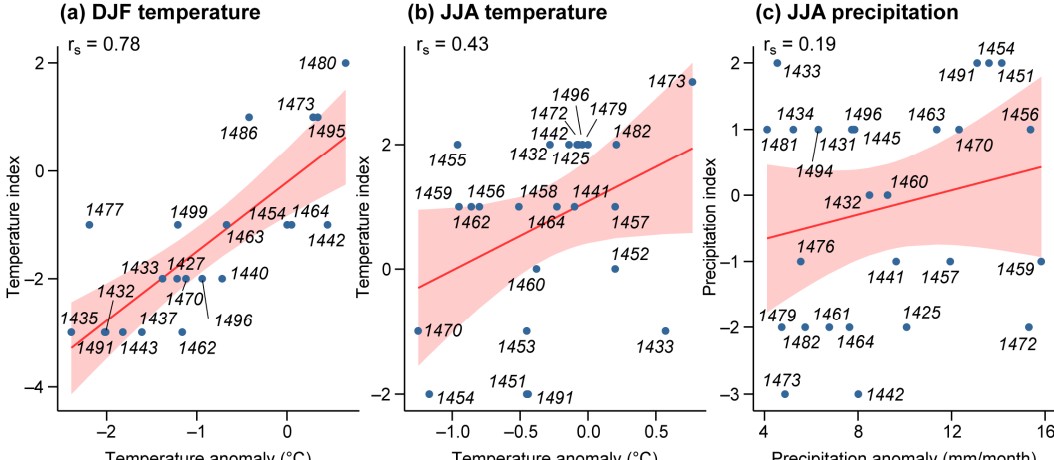

**Figure 10. Comparison of Czech DJF (a) and JJA (b) temperature indices and JJA precipitation indices (c) with corresponding temperature and precipitation anomalies from the ModE-RA reanalysis by Valler et al. (2024); $r_s$ – Spearman rank correlation.**

In the Czech Lands, the records kept by the chronicler Bartošek of Drahonice feature a particularly interesting phenomenon

(Goll, 1893). For between 13 and 22 July 1441, he recorded fogs and "smokes" in Bohemia, far beyond any living memory, when the morning and evening sun "looked like blood". Three centuries later, similar events were described in Bohemia, when "dry fog" and a red sun were observed in summer 1783 after the Lakagígar volcano eruption in Iceland (Písek and Brázdil, 2006; Brázdil et al., 2017b); this also occurred in other European countries (Stothers, 1996). Evidence of volcanic eruptions from Iceland indicates an eruption in or around the Hekla volcano, but for 1440 (Frímann, 2011).

Despite some uncertainties in the identification and timing of large volcanic eruptions, particularly in the 1450s (Bauch, 2017; Esper et al., 2017; Abbott et al., 2021), their cluster in Fig. 6b coincides well with temperature fluctuations in Europe. In JJA temperatures by Luterbacher et al. (2016), a significant cooling appeared in 1453 as a response to the unknown 1452/53 eruption (Fig. 8a). The cooler Czech summers in 1453–1454 identified in the documentary sources (cf. Fig. 2) followed this eruption. The volcanic cooling persisted for about the next 15 years, and its Northern Hemisphere extent was

demonstrated in several TRW proxy reconstructions (Esper et al., 2017), but it did not appear further in the Czech JJA temperature indices (cf. Fig. 2). Strange atmospheric phenomena visible all over Europe in September 1465 as the result of a volcanic dust veil, but dated to 1464/1465, were described by Bauch (2017). The persistence of the cold period may be related to another Southern Hemisphere eruption in 1457 or 1458 (Abbott et al., 2021). Both 1458 and 1459 are estimated to be well below the 15th-century mean in the temperature sum reconstruction by Torbenson et al. (2023), with 1458 being the

third coldest year in the period 1447–1499 (Fig. 4a) and occurring during a continuous positive trend in growing season temperature. Although there is little long-term agreement between the temperature sum estimates and the JJA temperature



reconstruction of Luterbacher et al. (2016), in part due to differences at multidecadal scales, the latter also displays considerable negative departures for 1458 and 1459 (–0.84 and –1.70 °C, respectively, for the closest grid point to our general study region). The spatial patterns of cold JJA in Europe for 1453 and 1459 according to ModE-RA data are shown

in Fig. 11. While in the first case the most intense cooling occurred in Northern Europe (Fig. 11a), in the second case it concentrated to the Iberian Peninsula and Southern Europe (Fig. 11b). The 1450s also coincide with the lowest point of negative NAO values (Fig. 6c; Cook et al., 2019). Conversely, the only major eruption of the 15th century identified in the Greenland ice cores occurred in 1477 (Toohey and Sigl, 2017).

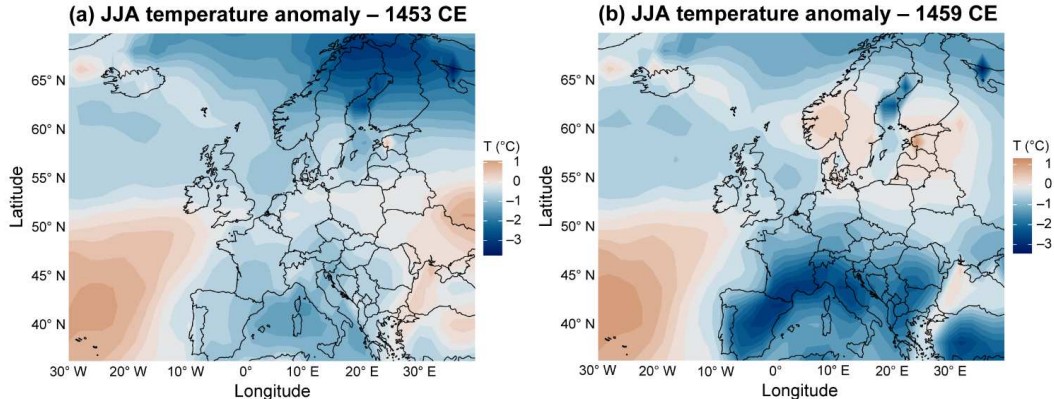

**Figure 11. Mean JJA temperatures in the years 1453 CE (a) and 1459 CE (b) in the Atlantic-European region, expressed as anomalies w.r.t. the 1961–1990 reference, according to the ModE-RA reanalysis (Valler et al., 2024).**

Czech documentary sources also indicate the occurrence of locusts in 1474–1475. An ancestor of Mikuláš Dačický of Heslov reported an incursion of locusts "*as long as a thumb, of a shiny greenish colour*, [with] *cattish, mask-like faces, their bellies like those of snakes*" that flew and clattered "*paying no heed to either people or livestock*" (Petrů and Pražák, 1955, p. 164;

Černá et al., 2018). This took place after the warm, dry patterns of 1473 in Europe (Camenisch et al., 2020). Before 1474–1475, locust outbreaks in the Czech Lands had been recorded in 1338 CE and after those dates in 1542–1546 (Brázdil et al., 2014). However, the latter contribution emphasises that locust outbreaks cannot be taken as clear indications of any particular climatic features in the Czech Lands at that time.

Although Camenisch (2017) investigated particularly famine and dearth during the 1480s and 1490s in Western and Central

Europe, the great famine of the early 1430s remains one of the most outstanding events of the 15th century in the Czech Lands (Brázdil et al., 2017a). The countryside and inhabitants were exhausted by the various negative consequences of the protracted Hussite wars, which had raged since the early 1420s. An already bad situation was heavily exacerbated by weather anomalies and HMEs. These included severe winters with deep frosts and large quantities of snow, periods of intense summer heat, droughts and/or devastating floods, leading to loss of lives and damage to property, to agriculture and

to crop production. The bad harvest in 1433 CE, with its accompanying shortages of grain and foodstuffs, led to price





increases, when a significant and rising number of the impoverished were forced to eat poor substitutes for their usual food (e.g. oak-leaves and acorns) in order to survive. This culminated in acute famine during the autumn (Höfler, 1865a; Goll, 1893). Extremely poor agroclimatic conditions for agricultural production in the early 1430s were confirmed by the reconstruction of agroclimatic zones, with particular focus on the eastern part of the Czech Lands, by Torbenson et al.

(2024), likely driven in part by low growing-season temperatures (see Fig. 4a). For example, the prices of rye and wheat in the country increased *c*. five-fold or six-fold in 1432 compared to those in 1431, from 5–6 Groschen to 30 Groschen (per 2.5 bushels). In 1434, the price of rye rose to an extraordinary 60 Groschen before the harvest (Brázdil and Kotyza, 1995). Wars and bad harvests aside, the high prices were also reflections of a continuing economic blockade inspired by religious animosity, and a decrease in the nominal value of the Czech Groschen (Čornej, 2000), as well as the speculation of greedy

merchants.

## 7 Conclusion

The main results of this analysis of the climate in the Czech Lands during the 15th century may be summarised as follows:

(i) The Old Czech Annals, chronicles, letters and the accountancy books for the town of Louny constitute the most important sources of information from documentary sources about the weather and related phenomena from the Czech Lands in the

15th century. The information thus originates almost exclusively in Bohemia. Secondary sources are of only limited use. Descriptions of the weather and related phenomena for individual years are relatively scarce, particularly for the 1400s and 1410s, and relatively the most frequent for the 1430s and 1450s.

(ii) Based on documentary data, available temperature and precipitation indices for the Czech Lands were interpreted on a 3-degree scale for months and on a 7-degree scale for seasons. Indices derived for winter and summer were the most frequent

compared with information from spring and autumn, which was sparse.

(iii) Czech temperature and precipitation indices accurately reflect the occurrence of extreme years and climate anomalies in Central Europe during the 15th century. Especially winter and, to some extent, summer Czech temperature indices highly correlate with the temperatures of the corresponding seasons obtained by the state-of-the-art paleo-reanalysis.

(iv) Especially the DJF temperature indices compiled from Czech documentary sources represent a unique source of

information about the variability of winter temperatures during the 15th century, about which natural proxies can provide very limited information. Documentary data offer insights on short-term events (such as the disastrous flood of 1432 during concurrent dry conditions) and non-growing season conditions (such as winter temperatures) that are impossible to extract from biological proxies.

(v) Continuous information concerning the climatic patterns of the Czech Lands during the 15th century may be obtained

only for temperature sum, precipitation totals and droughts reconstructed from fir and oak tree-ring chronologies.



(vi) Despite the incompleteness of the Czech indices, their great potential for the future can be seen in their combination with similar indices from other parts of Europe, as they can provide unique information especially about the spatial variability, extent, and intensity of climatic anomalies and HMEs in Europe in the 15th century.

**Appendix A**

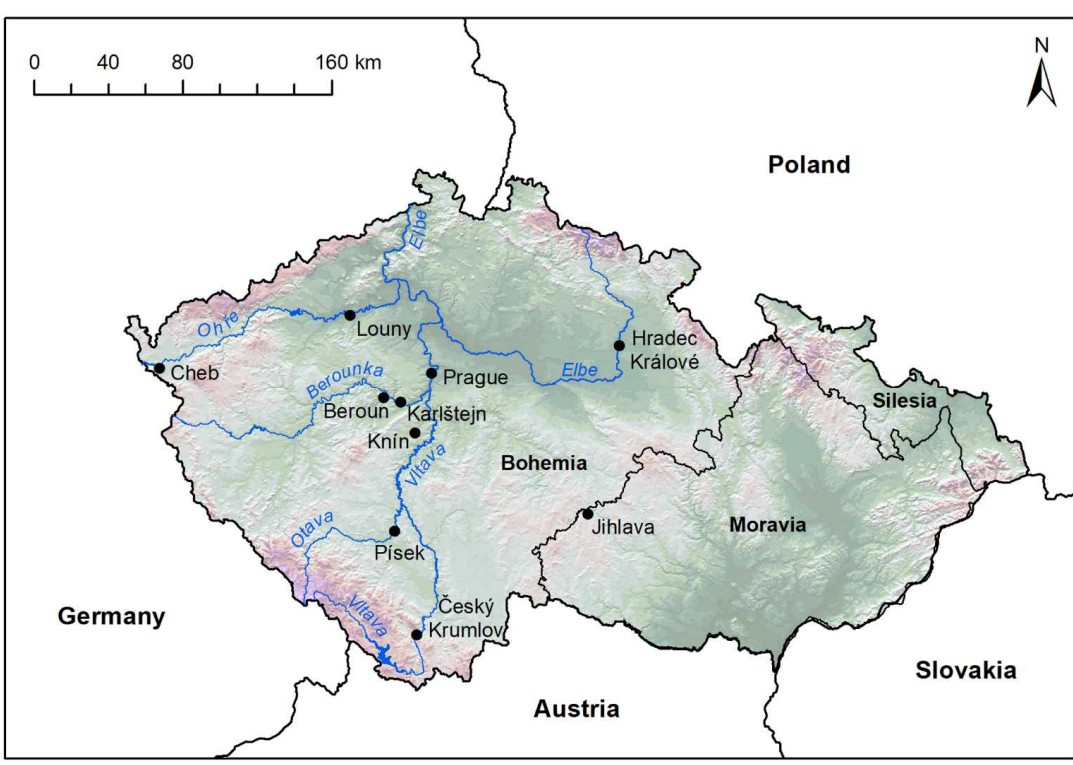


**Figure A1. Location of the places and rivers reported herein over the territory of the recent Czech Republic (data source: ArcCR 500 v2.0).**



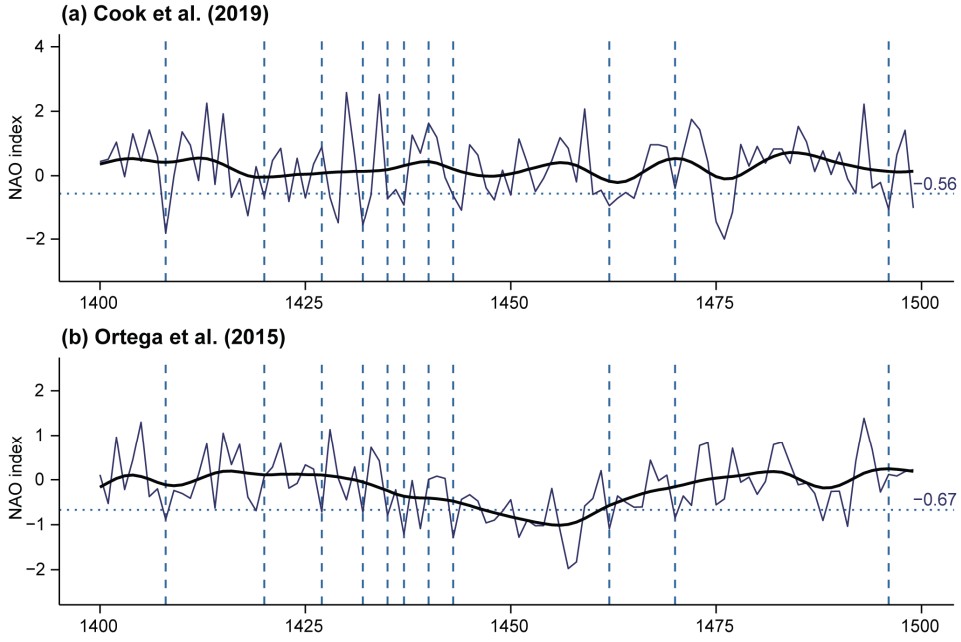

**Figure A2. North Atlantic Oscillation (NAO) reconstructions by (a) Cook et al. (2019) and (b) Ortega et al. (2015) for the 15th century. Smoothed by a 10-year Gaussian filter. Vertical dashed lines denote cold winters according to the Czech documentary indices. Horizontal dotted lines show the mean NAO index during the cold winters (–0.56 for Cook et al., 2019 and –0.67 for Ortega et al., 2015).**

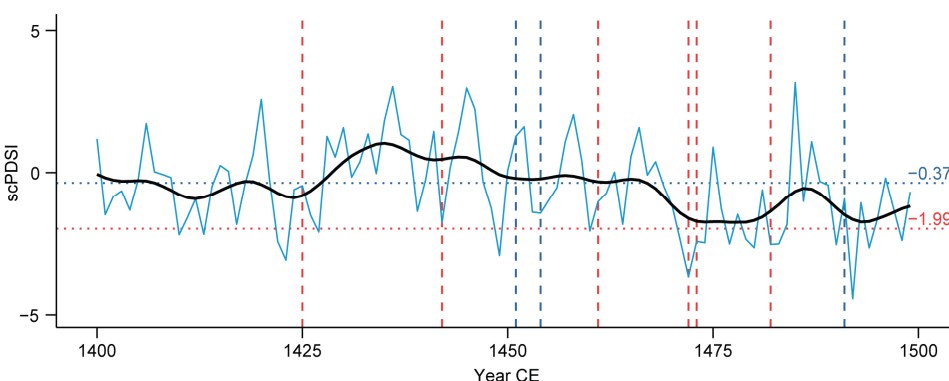



**Figure A3. Czech JJA scPDSI reconstruction for the 15th century by Büntgen et al. (2021). Smoothed by a 10-year Gaussian filter. Vertical dashed lines denote warm-dry (red) and cold-wet (blue) summers according to the Czech documentary indices. Horizontal dotted lines show the mean scPDSI during the warm-dry (red) and cold-wet (blue) summers (–1.99 and –0.37 respectively).**

**Data availability**

The basic Czech documentary data used in this article are available in Supplement and corresponding time series used are available on Brázdil, R., Dobrovolný, P., Torbenson, M. C. A., Dolák L., and Chromá, K.: 15th century climate in the Czech Lands and its Central European context, Zenodo [data set], https://doi.org/10.5281/zenodo.16268128, 2025.

**Supplement.** The supplement related to this article is available online at: https//doi.org/xxxxx.

**Author contributions**

RB interpreted documentary data, created temperature and precipitation indices, designed and wrote the paper with contributions from all co-authors. PD contributed with comparison of the Czech data with other climate reconstruction and analysed the 15th century climate in Central Europe. MCAT extended statistical analyses comparing Czech data with other proxy-based series and contributed to discussion. LD contributed to specification of Czech narrative weather sources and references. KC prepared the final version of figures and the whole manuscript for submission.

**Competing interests**

The contact author has declared that none of the authors has any competing interests.

**Acknowledgements**

We particularly acknowledge the work of our deceased colleague Oldřich Kotyza (Litoměřice), who excerpted weather-related data for the 15th century from various documentary sources, forming the basis for the preparation of this article. Tony Long (Carsphairn, Scotland) and Laughton Chandler (Charleston, SC) are acknowledged for English language corrections.

**Financial support**

This research was supported by the Johannes Amos Comenius Programme (P JAC), project No. CZ.02.01.01/00/22_008/0004605, Natural and anthropogenic georisks, and by the Czech Science Foundation, project no. 25-15855S, Internal climate variability over Europe. MCAT acknowledges support from the European Research Council (Advanced Grant #882727).

**Archival sources**



(AS1): Regionální muzeum Litoměřice, inv. č. SV 14142: Kalendář Historický. To jest krátké poznamenání všech dnuov jednoho každého měsíce přes celý rok. K nim přidány jsou některé paměti hodné Historiae o rozličných příhodách a proměnách, jak národuov jiných a zemí v Světě, tak také a obzvláštně národu i Království Českého z hodnověrných Kronik. S pilnosti sebráno, vytištěno a vydáno prací a nákladem M. Daniele Adama z Veleslavína. Vytlačeno v Starém Městě Pražském. Leta posledního věku: MDXC.

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
