# Peer review of "15th century climate in the Czech Lands and its Central European context"

_EGUsphere, 2025_

## Referee Comment (RC3)

**Review Brazdil et al 15th century by Christian Pfister**

The paper is innovative and convincing in terms of methodology and content

The results are attractively presented and well documented.

**Major improvements**

**Sect 3.1. Documentary data**

By including seasonal indices for Central Europe (Pfister, Wanner 2021) <a href="https://boris.unibe.ch/191962/">https://boris.unibe.ch/191962/</a> the number of missing indices might be reduced. This would be particularly crucial for winter., for which just the NAO study by Cook et al. 2019 is available.

Line 150: It would be worthwhile to compare the tree-ring reconstruction of the winter (December–March) of the North Atlantic Oscillation (NAO) by Cook et al. (2019) with the results to the 15th century winter indices for Central Europe <a href="https://boris.unibe.ch/191962">https://boris.unibe.ch/191962</a> b

Line 154-155 many indices are still missing. Most of the seasonal 15th century indices published in <a href="https://boris.unibe.ch/191962">https://boris.unibe.ch/191962</a> based on Pfister and Wanner 2021 refer to Germany. These results should also be considered.

Line 156-158 the dating of documentary sources according to the Julian style is prone to error, For clarity the term "Julian" or "Jul" should be added to the date or preferably the dates should be presented according to the Gregorian followed by abbreviation "Greg"

**Small modifications**

**Refereces**

Cook, E., Kushnir, Y., Smerdon, J., Williams, A., Anchukaitis, K., and Wahl, E.: A Euro-Mediterranean tree-ring 680 reconstruction of the winter NAO index since 910 C.E., Clim. Dyn., 53, 1567–1580, https://doi.org/10.1007/s00382-019-04696-2, 2019.----not in alphabetical order

---

## Referee Comment (RC4)

**15th century climate in the Czech Lands and its Central European context**

Rudolf Brázdil1,2, Petr Dobrovolný1,2, Max Carl Arne Torbenson2,3, Lukáš Dolák1,2, Kateřina Chromá2

1Institute of Geography, Masaryk University, Brno, Czech Republic

[referee-annotated manuscript omitted]

---

## Referee Comment (RC5)

**Comment on Macdonald on Brázdil et al. in review by Christian Pfistet**

Line 2 agreed Line 15-16 agreed Line 19 I think that information on winter is indeed unique corresponding quotations should be provided Lines 52 and 55 agreed Line 64 Line 65 agreed Line 77 are these details really needed? Line 84 Although religious wars had left the Czech Lands totally devastated by the end of the 15th century, Line 90 agreed keep the sentence Line 127 Line 167 add reference Dobrovolny et al 2010 Line 231 agreed Line 248 agreed Line 318 agreed Linw421 add severity Line 437 this is an interesting point. Perhaps the Central European indices by Pfister would be of use

Line 504

agreed

---

## Author Comment (AC1)

This is a very comprehensive work that develops knowledge on a topic or area that was lacking in its entirety regarding the climate of the Late Middle Ages in Central Europe. The authors express the limitations and challenges of the availability of sources, which demonstrates their expertise and honesty. The analysis of the information obtained is optimal, and the results are integrated into the context, which the authors themselves summarize very accurately.

RESPONSE: We would like to thank the anonymous referee #1 for generally positive evaluation of our study as well as several useful comments, which we are responding to below.

I have no general criticisms to raise, but only some minor details that I leave for the authors' consideration if they would introduce or consider some of the suggestions:

+ Section 3. "Documentary data." Would it be more appropriate to express this as "Documentary Sources"?

RESPONSE: Accepted and changed as "Documentary sources".

+ At various times, the difficulty in finding information to cover all the years under study within the 15th century is explained. Don't the authors consider creating groupings by 5 or 10 years to overcome this problem? At least in some cases, as a support for the annual study, continuous diagrams by groupings would perhaps provide a complementary result. RESPONSE: We do not believe that grouping data by 5 or 10 years would be able to overcome the problem of missing information. To only work with years for which we have documentary data seems to us to be more scientifically correct, because limited information is not always representative enough for 5- or 10-year intervals.

A major advantage of documentary records is the unrivalled temporal resolution for information on pre-20th century conditions. The comparisons with biological proxy data presented highlight this in some cases, as tree growth (or the processes that result in the isotopic composition of the annual wood layer) record climate over longer (i.e., seasonal) temporal windows than some of the documentary events (e.g., frost events or torrential rain). By averaging these types of data at sub-decadal or decadal scales, there is an obvious loss of information. Because the central aim of our manuscript is to collate and introduce this unique material, these longer averages and comparisons could be considered secondary. However, we do agree that the approach can be very useful in other instances, as evident by previous publications by some of the authors (e.g., Brázdil et al. 2013). Reference:

Brázdil, R., Dobrovolný, P., Trnka, M., Kotyza, O., Řezníčková, L., Valášek, H., Zahradníček, P., and Štěpánek, P.: Droughts in the Czech Lands, 1090–2012 AD, Clim. Past, 9, 1985–2002, https://doi.org/10.5194/cp-9-1985-2013, 2013.

+ Fig. 2. Displaying a time axis in successive units but without maintaining its consecutive timeline of years creates problems in interpreting the information. There are jumps or gaps that cannot be perceived, and it seems to be a continuous series when in reality it is not. Wouldn't it be possible to present the information with axes that correctly visualize the chronological progression? For example, by marking the years without information with a softer colour or gray colour?, without breaking the continuity of the annual series. RESPONSE: Accepted, we prepared the new version of Fig. 2 showing continuously missing data or available indices for the whole 15th century – see below:

+ Lines 305-320. The level of disagreement between the dendroclimatic and historical data is explained. Could this low level of consistency be the result of the different geographic locations of the two proxies? Could this be explained in the text? I know of countries where these comparisons have been attempted, but the areas where the dendroclimatic and historical data are obtained are completely different, with their own ecosystems and dynamics. Therefore, the differences are entirely logical, even considering that the objective is to assess climate variability on a broad temporal scale. I don't think these differences imply the slightest loss or relativization of the quality of the results.

RESPONSE: We fully agree with the overall sentiment of the reviewer's comment, and we touched upon this briefly on lines 314-320, including: "These varying relationships may reflect the fact that precipitation extremes tended to be spatially restricted to the extent that

they found no reflection in documentary sources. Further, TRW-based precipitation reconstructions display relatively low skill. Finally, some extreme years or seasons are not reflected in precipitation indices simply because of the low density of available documentary evidence."

Beyond this, we also believe that the temporal resolution (part of "their own ecosystem and dynamics") that we referenced to in the above comment on decadal averaging likely plays a role here. Despite all these complexities, there are statistically significant relationships between the different data sources – which in itself is remarkable. We have modified the wording to reinforce this view in paragraph before Table2 (lines 316-318) as follows: "As follows from Table 2, full or only partial agreement appears for just a few of them. These varying relationships may reflect the fact that precipitation extremes tended to be spatially heterogenous to the extent that they found no reflection in documentary sources, or potentially due to differences in the resolution of recording (i.e., the biological proxies tend to incorporate conditions over the full growing season and often fail to capture events at submonthly scales). Further, TRW-based precipitation reconstructions display relatively low skill."

Hopefully the revised framing has clarified the issue.

+ Line 487, p. 24. A strong volcanic eruption from 1452-1453 is mentioned as "unknown," but perhaps this isn't the eruption of Mount Kuwae?

RESPONSE: We do not believe that there is an adequate consensus on the topic of origin (Ballard et al., 2023). It is worth noting that the 1452/53 event is more prominent in Northern Hemisphere records, and it has therefore been suggested that it was the result of an extratropical eruption (Burke et al., 2023). The Kuwae event, which previously was placed in 1452/53 (Gao et al., 2006), is thought by some to have originated in 1458/59 (e.g., Gautier et al., 2021) – which has, in part, been corroborated very recently by Antarctic ice core data (Ro et al., 2025). This latter study by Ro et al. (2025) also raises the possibility of two independent but concurrent eruptions around 1458 – making the story even more complex. As we are not tephrochronologists, we prefer to not attribute the eruption(s) directly (and doing so/not doing so does not change the story of our manuscript). Nonetheless, we tried slightly change the corresponding paragraph (lines 485-493) as follows:

"Despite some uncertainties in the identification and timing of large volcanic eruptions, particularly in th 1450s (Bauch, 2017; Esper et al., 2017; Abbott et al., 2021; Ro et al., 2025), their cluster in Fig. 6b coincides well with temperature fluctuations in Europe. In JJA temperatures by Luterbacher et al. (2016) (Fig. 8a), a significant cooling appeared in 1453 as a response to an eruption of debated origins in 1452/53 (□g., Ballard □t al., 2023; Burk□□t al., 2023), previously attributed by Gao et al. (2006) to the Kuwae volcano eruption in Vanuatu. The cooler Czech summers in 1453–1454 identified in the documentary sources (cf. Fig. 2) followed this eruption. The volcanic cooling persisted for about the next 15 years, and its Northern Hemisphere extent was demonstrated in several TRW proxy reconstructions (Esper et al., 2017), but it did not appear further in the Czech JJA temperature indices (cf. Fig. 2). Strange atmospheric phenomena visible all over Europe in September 1465 as the result of a volcanic dust veil, but dated to 1464/1465, were described by Bauch (2017). The persistence of the cold period may be related to another Southern Hemisphere eruption in 1457 or 1458 (Abbott et al., 2021). Moreover, the recent analysis of Antarctic ice core data by Ro et al. (2025) mentions the possibility of two independent but concurrent eruptions around 1458." New references:

Ballard, C., Bedford, S., Cronin, S. J., and Stern, S.: Evidence at source for the mid-fifteenth century eruption of Kuwae, Vanuatu, J. Appl. Volcanol., 12, 12, https://doi.org/10.1186/s13617-023-00138-1, 2023.

Burke, A., Innes, H. M., Crick, L., Anchukaitis, K. J., Byrne, M. P., Hutchison, W., McConnell, J. R., Moore, K. A., Rae, J. W. B., Sigl, M., and Wilson, R.: High sensitivity of summer temperatures to stratospheric sulfur loading from volcanoes in the Northern Hemisphere, Proc. Natl. Acad. Sci. USA, https://doi.org/10.1073/pnas.2221810120, 2023. Gao, C., Robock, A., Self, S., Witter, J. B., Steffenson, J. P., Clausen, H. B., Siggaard-Andersen, M.-L., Johnsen, S., Mayewski, P. A. and Ammann, C.: The 1452 or 1453 A.D. Kuwae eruption signal derived from multiple ice core records: Greatest volcanic sulfate event of the past 700 years, J. Geophys. Res., 111, D12107, https://doi.org/10.1029/2005JD006710, 2006.

Ro, S., Hur, S. D., Ekaykin, A., Han, Y., Ro, C.-U., Hong, S.-B., Lee, M. J., Chang, C., Lee, S., Moon, J., Jung, H., Veres, A., Lee, A., and Hong, S.: Origin of the 1458/59 CE volcanic eruption revealed through analysis of glass shards in the firn core from Antarctic Vostok station, Commun. Earth Environ., 6, 828, https://doi.org/10.1038/s43247-025-02797-x, 2025.

---

## Author Comment (AC2)

The article is very interesting and valuable, providing a lot of important information about the weather and climate in the Czech Lands in the 15th century. That is why I suggest its publication in the journal *Climate of the Past*; however, it still requires some corrections and clarifications.

RESPONSE: We would like to thank the anonymous referee #2 for generally positive evaluation of our study as well as several comments, which we are trying respond to below.

**Main weaknesses:**

1. In the Introduction part, I suggest including a short summary of the present state of the art knowledge about 15th-century weather and climate in the Czech Lands (now the Czech Republic).

RESPONSE: In lines 41-44 we mention following: "However, surviving Czech documentary evidence before the 16th century is sparser, permitting description of only some weather/climatic patterns and HMEs in certain years (Brázdil and Kotyza, 1995, 1997), which were used, for example, by Brázdil et al. (2017a) to analyse severe famines in the 1280s, 1310s, and the early 1430s in the Czech Lands in relation to weather and climate conditions." We hope, that here we characterise what is available until now for the 15th century in the Czech Lands and what was done, i.e. "description of only some weather/climatic patterns and HMEs in certain years" and their further use for the analysis of "severe famines in the 1280s, 1310s, and the early 1430s".

2. I suggest significantly shortening Section 2, as it is definitely too long and detailed, and is only loosely connected with the main subject of the paper, although it gives some background on the political, social and economic situation.

RESPONSE: The description of the political, social and economic situation in the Czech Lands during the 15th century only encompasses 39 lines from the total 553 lines of the manuscript. We believe this to be necessary to understand the many factors that undoubtedly influenced the availability and number of documentary sources, from which weather/climatic reports were derived. We believe, that in historical climatology papers we should at least partly (as here) take in account related historical situation for complex evaluation of the analysed topic (see e.g. lines 514–530, where historical context is necessary). Moreover, another referee Dr. Neil Macdonald commented this chapter as follows: "A good section that explains why the different sources are found, and the socio-economic context to the records, which is often over looked." These are reasons, from which we would like to preserve Sect. 2 in its original extent, because Climate of the Past has readers not only among climatologists.

3. The Discussion part can be improved by comparison of the presented results for Czech Lands with the available weather and climate information (many quantitative reconstructions based on both documentary evidence and natural proxies) from the neighbouring area of Poland (see Ghazi et al. 2023, https://doi.org/10.1016/j.jhydrol.2023.129778; Przybylak et al. 2023, https://doi.org/10.5194/cp-19-2389-2023, the last item is cited in the paper, only generally), because, as result from the reviewed paper as well as Luterbacher et al. (2010), both areas are very well correlated, in particular in case of winter air temperature. Also, because both areas are included in Central Europe, in the case of the ModE-RA paleoreanalysis data used in the paper. It should also be remembered that the 15th-century reconstructions from ModE-RA are entirely modelled data without any assimilation of data from this region. For Europe, only one series of data

from the Low Countries (Van Engelen et al., 2003, see https://mode-ra.unibe.ch/climeapp/) was used for the period from October to March. The situation is better for the warm half-year, for which proxy data (mainly tree-ring widths) are mainly available from SW Europe and Fennoscandia; however, no data are available from Central Europe, the entire Eastern Europe, and SE Europe.

Luterbacher J., Xoplaki E., Küttel M., Zorita E., González-Rouco J. F., Jones P. D., Stössel M., Rutishauser T., Wanner H., Wibig J., Przybylak R., 2010, Climate Change in Poland in the Past Centuries and Its Relationship to European Climate: Evidence From Reconstructions and Coupled Climate Models. In: Przybylak R, Majorowicz J, Brázdil R, Kejna M (eds) The Polish Climate in the European Context: An Historical Overview, Springer, Berlin Heidelberg New York, 3-39.

RESPONSE: Following the referee's comment on the matter of Polish papers, we added the proposed papers and their results into the first paragraph of Sect. 6 Discussion as follows: "Many of the weather events and anomalies, as well as the derived temperature/precipitation indices, reported herein for the Czech Lands were also documented in other European regions or countries, such as Germany (Glaser, 2008), the Low Countries (van Engelen et al., 2001, 2009), the Burgundian Low Countries (Camenisch 2015), Poland (Przybylak et al., 2023) and the western-central European area (Pfister and Wanner, 2021). Interpretation of a number of severe winters, extending to March or April in the Czech Lands during the 1430s (cf. Fig. 2), together with the occurrence of floods (also, in part, windstorms and convective storms), confirm the severe character of the cold 1430s in Europe, as described by Camenisch et al. (2016). Similarly for Poland, Przybylak et al. (2023) mentioned a higher frequency of cold and very cold winters for the 1430s as-well as their higher wetness (cf. Fig. 2). Another similarity between the Czech Lands and Poland concerning of moisture regime is dictated mainly by summer indices. Pfister et al. (2024), analysing wine must quality as a reflection of weather patterns for Germany, Luxembourg, eastern France, and the Swiss Plateau in 1420– 2019 CE, identified the years 1470–1479 as having the highest average quality on the decadal scale, which is correlating well with warm summers in the 1470s in Poland (Przybylak et al., 2023). On the other hand, years of poor wine quality in 1453–1466 and 1485–1494 were attributed to prevailingly cold and wet summers (Pfister et al., 2024). In the subsequent paper dealing with wine must yields for 1416-1988 CE, Pfister et al. (2025) identified as "good harvest" years those between 1416 and 1425 and further 1471–1473. Of the years of drought and low water levels in medieval Hungary (Kiss and Nikolić, 2015; Kiss, 2017), dry patterns in the Czech Lands tallied with those that they highlight in 1473, 1479 and 1482. The year of 1455, with a warm summer, was probably dry there too. Concerning floods as an opposite extreme to droughts, from identified 18 flood years in the Czech Lands 14 such years (i.e., 77.8%) agreed with flood years selected for Poland by Ghazi et al. (2023)."

Concerning of comparison with other indices as proposed by the referee, we included a new paragraph connected to Fig. 10 as follows:

"Quantitative verification of Czech indices for DJF and JJA temperatures and JJA precipitation used in Fig. 10 can be also performed for temperature and precipitation indices from the Low Countries (van Engelen et al., 2001, 2009). Although the temperature indices were defined on different scale (from 1 to 9) and for differently defined seasons (November to March for DJF and May to September for JJA patterns), they show strong and statistically significant (p <0.05) Spearman rank correlation with the Czech indices particularly for DJF temperatures (0.89) and JJA precipitation (0.83) and naturally slightly lower correlation for JJA temperatures (0.54). Comparison of Czech temperature indices with those derived for Western and Central Europe by Pfister and Wanner (2021) gives a lower correlation for DJF temperatures (0.75) and a higher correlation for JJA temperatures (0.62) than with Low

Countries, but statistically significant in both cases (p <0.05). Much smaller number of JJA precipitation indices did not allow to compare both considered datasets.

As for above comparison of Czech with van Engelen et al. (2001, 2009) and Pfister and Wanner (2021) indices is necessary to note, that temperature indices from both datasets show clearly higher frequency of very cold and extremely cold DJFs than of very warm and extremely warm DJFs. Similarly, there appeared also significantly higher number of hot and extremely hot JJAs compared to very cold and extremely cold JJAs. However, this feature does not reflect properly climatic patterns of the 15th century, but it rather points out to a specific extreme-oriented feature of documentary indices (Brázdil et al., 2005)."

As for the referee's point It should also be remembered that the 15th-century reconstructions from ModE-RA are entirely modelled data without any assimilation of data from this region we add that we are aware of the limitations of ModE-RA data for the 15th century. We agree that the resulting reanalysis is primarily defined by the results of the model used especially for winter. For this reason, we also provide the SD ratio (see Sect. 4 Methods, lines 176–179). In Sect. 5.4 we point out that high SD ratio values indicate generally higher uncertainties associated with the use of ModE-RA data. Nevertheless, we believe that the reanalysis is a valuable source of information on the climate of a significant part of the 15th century, as it provides physically consistent estimates of several climate variables at monthly resolution. Furthermore, the added value of this data source, e.g., for analysis of past hydrometeorological extremes, has been demonstrated in several studies (e.g., Valler et al., 2024; Brönnimann et al., 2025).

**New references:**

Brönnimann, S., Franke, J., Valler, V., Hand, R., Samakinwa, E., Lundstad, E., Burgdorf, A.-M., Lipfert, L., Pfister, L., Imfeld, N., and Rohrer, M.: Past hydroclimate extremes in Europe driven by Atlantic jet stream and recurrent weather patterns, Nat. Geosci., 18, 246–253, https://doi.org/10.1038/s41561-025-01654-y, 2025.

Ghazi, B., Przybylak, R., Oliński, P., Bogdańska, K., and Pospieszyńska, A.: The frequency, intensity, and origin of floods in Poland in the 11th–15th centuries based on documentary evidence, J. Hydrol., 623, 129778, https://doi.org/10.1016/j.jhydrol.2023.129778, 2023. van Engelen, A. F. V., Ijnsen, F., Buisman, J., and van der Schrier, G.: Precipitation indices Low Countries, in: Poster Abstracts of the Millennium Milestone Meeting 3, edited by: Young, G. and McCarroll, D., Cala Millor, Mallorca, 62–63, 2009.

**Minor weaknesses**

1. lines 176-177 – not clear the area for which SD was calculated, Central Europe, or a smaller area encompassing only the Czech Lands?,

RESPONSE: The corresponding sentence was corrected as follows: "Using the ClimeApp application (Warren et al., 2024), we calculated for Central Europe (45–55° N, 5–25° E) the standard deviation (SD) ratio, which helps clarify the differences found from the above tests using ModE-RA reanalysis."

2. lines 458-460 – I suggest rewriting these sentences slightly, taking into account the information given at the end in point 3 (Major weaknesses),

RESPONSE: We wrote in the cited lines following text: "Furthermore, this is completely independent source in this study, as no data from the Czech Lands prior to 1500 CE has been assimilated in ModE-RA dataset. One disadvantage is that the density of different types of proxies is relatively low in the 15th century and the re-analysis is dominated by the ensemble mean of the atmospheric circulation model in this period (see Hand et al., 2023 and Valler et

al., 2024 for more details)." We believe that nothing is wrong in our statements and we do not know what kind of a slight correction the reviewer expects.

3. lines 487-488 - this is a very well-known volcano eruption (Kuwae in Vanuatu), only the precise date is not established yet. The most probable date is 1452/1453 CE; however, in literature, other dates are also given, most of which fall in the 1450s. Kuwae was one of the largest eruptions in the past millennium,

RESPONSE: We do not believe that there is an adequate consensus on the topic of origin (Ballard et al., 2023). It is worth noting that the 1452/53 event is more prominent in Northern Hemisphere records, and it has therefore been suggested that it was the result of an extratropical eruption (Burke et al., 2023). The Kuwae event, which previously was placed in 1452/53 (Gao et al., 2006), is thought by some to have originated in 1458/59 (e.g., Gautier et al., 2021) – which has, in part, been corroborated very recently by Antarctic ice core data (Ro et al., 2025). This latter study by Ro et al. (2025) also raises the possibility of two independent but concurrent eruptions around 1458 – making the story even more complex. As we are not tephrochronologists, we prefer to not attribute the eruption(s) directly (and doing so/not doing so does not change the story of our manuscript). Nonetheless, we tried slightly change the corresponding paragraph (lines 485-493) as follows:

"Despite some uncertainties in the identification and timing of large volcanic eruptions, particularly in the 1450s (Bauch, 2017; Esper et al., 2017; Abbott et al., 2021; Ro et al., 2025), their cluster in Fig. 6b coincides well with temperature fluctuations in Europe. In JJA temperatures by Luterbacher et al. (2016) (Fig. 8a), a significant cooling appeared in 1453 as a response to an eruption of debated origins in 1452/53 (e.g., Ballard et al., 2023; Burke et al., 2023), previously attributed by Gao et al. (2006) to the Kuwae volcano eruption in Vanuatu. The cooler Czech summers in 1453–1454 identified in the documentary sources (cf. Fig. 2) followed this eruption. The volcanic cooling persisted for about the next 15 years, and its Northern Hemisphere extent was demonstrated in several TRW proxy reconstructions (Esper et al., 2017), but it did not appear further in the Czech JJA temperature indices (cf. Fig. 2). Strange atmospheric phenomena visible all over Europe in September 1465 as the result of a volcanic dust veil, but dated to 1464/1465, were described by Bauch (2017). The persistence of the cold period may be related to another Southern Hemisphere eruption in 1457 or 1458 (Abbott et al., 2021). Moreover, the recent analysis of Antarctic ice core data by Ro et al. (2025) mentions the possibility of two independent but concurrent eruptions around 1458." New references:

Ballard, C., Bedford, S., Cronin, S. J., and Stern, S.: Evidence at source for the mid-fifteenth century eruption of Kuwae, Vanuatu, J. Appl. Volcanol., 12, 12, https://doi.org/10.1186/s13617-023-00138-1, 2023.

Burke, A., Innes, H. M., Crick, L., Anchukaitis, K. J., Byrne, M. P., Hutchison, W., McConnell, J. R., Moore, K. A., Rae, J. W. B., Sigl, M., and Wilson, R.: High sensitivity of summer temperatures to stratospheric sulfur loading from volcanoes in the Northern Hemisphere, Proc. Natl. Acad. Sci. USA, https://doi.org/10.1073/pnas.2221810120, 2023. Gao, C., Robock, A., Self, S., Witter, J. B., Steffenson, J. P., Clausen, H. B., Siggaard-Andersen, M.-L., Johnsen, S., Mayewski, P. A. and Ammann, C.: The 1452 or 1453 A.D. Kuwae eruption signal derived from multiple ice core records: Greatest volcanic sulfate event of the past 700 years, J. Geophys. Res., 111, D12107, https://doi.org/10.1029/2005JD006710, 2006.

Ro, S., Hur, S. D., Ekaykin, A., Han, Y., Ro, C.-U., Hong, S.-B., Lee, M. J., Chang, C., Lee, S., Moon, J., Jung, H., Veres, A., Lee, A., and Hong, S.: Origin of the 1458/59 CE volcanic eruption revealed through analysis of glass shards in the firn core from Antarctic Vostok station, Commun. Earth Environ., 6, 828, https://doi.org/10.1038/s43247-025-02797-x, 2025.

4. lines 492-493 - it seems that that eruption can also be attributed to the Kuwae volcano; see what Abbott et al. (2021) wrote in the Introduction part of the cited paper: 'The large sulfate-loading eruption during the 1450s CE has most commonly been attributed to the formation of the submarine Kuwae caldera offshore of Vanuatu in the South Pacific.'

RESPONSE: Our expression in lines 492-493 says: "The persistence of the cold period may be related to another Southern Hemisphere eruption in 1457 or 1458 (Abbott et al., 2021)." We are just citing what is presented in Abbott et al. (2021) on page 565, i.e. we do not see to join it with the Kuwae volcano. Otherwise please see our expression to the preceding point 3.

---

## Author Comment (AC3)

**Review Brazdil et al 15th century by Christian Pfister**

The paper is innovative and convincing in terms of methodology and content. The results are attractively presented and well documented.

RESPONSE: We would like to thank Christian Pfister for generally positive evaluation of our study as well as several comments, which we are trying to respond below.

**Major improvements**

Sect 3.1. Documentary data

By including seasonal indices for Central Europe (Pfister, Wanner 2021) https://boris.unibe.ch/191962/ the number of missing indices might be reduced. This would be particularly crucial for winter., for which just the NAO study by Cook et al. 2019 is available. RESPONSE: The aim of our article is formulated as follows: "The aim of this contribution is to address research gaps concerning the 15th century in the Czech Lands by presenting the existing knowledge related to weather/climate and HMEs from available documentary evidence. The analysis concentrates on climate variability expressed by temperature and precipitation indices, documented HMEs, and comparison of these results with other climate reconstructions and data sources from Central Europe." We present Czech temperature and precipitation indices only for cases, in which the corresponding Czech documentary evidence is available and can be used to interpret corresponding indices. Our intention is not to develop any "artificial" Czech indices that also cover years with no Czech documentary sources and being derived from other data sources, for example like those presented on https://boris.unibe.ch/191962/. Nonetheless, for years with available Czech indices we did comparison with Pfister and Wanner (2021) indices – see our response to lines 154-155 below.

Line 150: It would be worthwhile to compare the tree-ring reconstruction of the winter (December–March) of the North Atlantic Oscillation (NAO) by Cook et al. (2019) with the results to the 15th century winter indices for Central Europe https://boris.unibe.ch/191962. RESPONSE: Please see preceding explanation of the aim of our study. We are not working in the scale of Central Europe, but we only compare our "results with other climate reconstructions and data sources from Central Europe."

Line 154-155 many indices are still missing. Most of the seasonal 15th century indices published in https://boris.unibe.ch/191962 based on Pfister and Wanner 2021 refer to Germany. These results should also be considered. RESPONSE: As explained above, complementing of missing indices is out of intention of this article. To consider comparison of Czech indices with those from Pfister and Wanner (2021), we added a new paragraph in Sect. 6 Discussion, below the Fig. 10 as follows: "Quantitative verification of Czech indices for DJF and JJA temperatures and JJA precipitation used in Fig. 10 can be also performed for temperature and precipitation indices from the Low Countries (van Engelen et al., 2001, 2009). Although the temperature indices were defined on different scale (from 1 to 9 degree) and for differently defined seasons (November to March for DJF and May to September for JJA patterns), they show strong and statistically significant (p < 0.05) Spearman rank correlation with the Czech indices particularly for DJF temperatures (0.89) and JJA precipitation (0.83) and naturally slightly lower correlation for JJA temperatures (0.54). Comparison of Czech temperature indices with those derived for Western and Central Europe by Pfister and Wanner (2021) gives a lower correlation for DJF temperatures (0.75) and a higher correlation for JJA temperatures (0.62)

than with Low Countries, but statistically significant in both cases (p <0.05). Much smaller number of JJA precipitation indices did not allow to compare both considered datasets. As for above comparison of Czech with van Engelen et al. (2001, 2009) and Pfister and Wanner (2021) indices is necessary to note, that temperature indices from both datasets show clearly higher frequency of very cold and extremely cold DJFs than of very warm and extremely warm DJFs. Similarly, there appeared also significantly higher number of hot and extremely hot JJAs compared to very cold and extremely cold JJAs. However, this feature does not reflect properly climatic patterns of the 15th century, but it rather points out to a specific extreme-oriented feature of documentary indices (Brázdil et al., 2005)."

Line 156-158 the dating of documentary sources according to the Julian style is prone to error, For clarity the term "Julian" or "Jul" should be added to the date or preferably the dates should be presented according to the Gregorian followed by abbreviation "Greg" RESPONSE: Because of sentence "With respect to comparability with recent climate, the data presented were recalculated from the Julian calendar to the current Gregorian style by adding nine days to the original dates." on lines 156-157 is clear, that dating of events in the following text is made in the Gregorian style, i.e. there is not necessary to use the abbreviation "Greg", which is usually not used in publications. Before this statement an exact dating was used only in Sect. 3.1, point (i), where we corrected it as "[30 November 1434, Julian calendar]" and in point (ii), where we changed it as "on the Thursday [14 July, Julian calendar]". All other dates in Sect. 3.1 are already recalculated to the Gregorian calendar.

**Small modifications**

References

Cook, E., Kushnir, Y., Smerdon, J., Williams, A., Anchukaitis, K., and Wahl, E.: A Euro-Mediterranean tree-ring 680 reconstruction of the winter NAO index since 910 C.E., Clim. Dyn., 53, 1567–1580, https://doi.org/10.1007/s00382-019-04696-2, 2019.-----not in alphabetical order

RESPONSE: According to the Czech alphabetical order our ordering is valid  $(A, B, C, \check{C}, D, ...)$ . If there is not distinguished between C and  $\check{C}$  in the English alphabet, then the referee's comment is correct, and we will make the change.

---

## Author Comment (AC4)

**Referee #4**

This is a well-developed paper that presents new novel precipitation and temperature reconstructions for the Czech Lands. The attached manuscript offers some thoughts and comments; however, these are almost all stylistic and structural rather than substantive issues that need to be addressed. I hope the authors find these helpful in reviewing the manuscript. I enjoyed reading the paper, thank you.

RESPONSE: We would like to thank Neil Macdonald for the generally positive evaluation of our study as well as for all thoughts and comments in the manuscript, which we are trying to respond below.

Line 10:

Response: The sentence "Secondary sources are only of limited use." was deleted.

Lines 15-16:

Response: The sentence was corrected as proposed, i.e.: "These indices are more frequent for winter and summer, with fewer indices derived for spring and autumn."

Line 19:

RESPONSE: Corrected as: "provide valuable information"

Section 2: A good section that explains why the different sources are found, and the socio-economic context to the records, which is often over looked.

RESPONSE: Thank you.

Line 52: "far as the 1380s. Economic" - add a citation so the reader can follow up on these themes?

RESPONSE: The corresponding citation is on line 57 (i.e., Čornej, 2003).

Line 53:

RESPONSE: Corrected as: "cause a deep"

Line 55: "perceived as corrupt and inept. Inevitably" - add citation

RESPONSE: The corresponding citation is on line 57 (i.e., Čornej, 2003).

Line 64: move so reads 'largely Catholic Czech nobility'. assuming this is what you mean "the Czech nobility, largely Catholic."

RESPONSE: Corrected as: "largely Catholic Czech nobility"

Line 65:

RESPONSE: Corrected as: "to the isolation"

Lines 76-77: Proposed "... Pope Pius II, Jiri was excommunicated and his deposition as King of Bohemia was declared by the Catholic Church.

RESPONSE: We prefer our original expression with a small change: "with Pope Pius II, Jiří of Poděbrady was deposed by the Catholic Church in 1466 and cast into anathema."

Lines 80-82:

RESPONSE: The sentence was modified as proposed, i.e. "The shared kingship and the acute risk of the Czech Lands dividing ended with Matthias' death in 1490, the Czech Crown again unified."

Lines 83-84:

RESPONSE: The sentence was corrected as: "The religious wars had left the Czech Lands totally devastated by the end of the 15th century (Macek, 2001)."

Line 90:

RESPONSE: Corrected as: "3.1 Documentary sources"

Line 114: This is a good point, I am sure that there are probably lots of accounts of the weather from around Europe in the Vatican archives from papal emissaries over the last millenia.

RESPONSE: Thank you, we agree.

Lines 166-167: Worth adding a sentence here on studies that have demonstrated the value of this approach with overlapping instrumental series - high similarities can be achieved, however extremes tend to be missed?

RESPONSE: We have added this into the second paragraph in Sect. 6 Discussion as follows: "In the case of numerous proxy reconstructions based on natural archives, direct comparison is severely limited by their seasonality, which is often restricted to late MAM and JJA (see Fig. 8). Moreover, depending on the used calibration method, proxy reconstructions may underestimate the intensity of extremes (see McCarroll et al., 2015 or Možný et al., 2016a for the Czech Lands)."

New reference:

McCarroll, D., Young, G., and Loader, N.: Measuring the skill of variance-scaled climate reconstructions and a test for the capture of extremes, Holocene, 25, 618–626, https://doi.org/10.1177/0959683614565956, 2015.

Line 231: define here

RESPONSE: Corrected as: "records of hydrometeorological extremes (HMEs) and"

Line 248: add citation to "destroyed houses. Many"

RESPONSE: Citations to this and following descriptions are given on lines 253-254.

Line 310: "the positive scPDSI value" - I think this is also a good argument for potentially using different drought indices e.g. SPI derived from temperature would be an interesting comparison, does it show the same variability? higher signal to the dendro? RESPONSE: We fully agree that PDSI is not always the optimal target for dendroclimatological reconstruction. Because PDSI integrates both input (i.e., precipitation) and output (i.e., evaporation/evapotranspiration demands driven by temperature), it essentially becomes impossible to assert if a high (positive) PDSI value was caused by heavy precipitation or cold conditions. The Torbenson et al. (2023) paper used in the comparisons attempted to address this issue specifically, using the two predictors (carbon and oxygen stable isotopes from living and subfossil Czech oak trees) previously combined for the Büntgen et al. (2021) scPDSI reconstruction to separate temperature from water balance (the latter being very closely related to SPI). The results indicate that notable periods of high PDSI in pre-history were driven by vastly different hydrothermal conditions. For example, the exceptionally wet periods during the early 3rd century and late 4th century CE in the Büntgen

reconstructions are suggested to have stemmed from almost opposite drivers (i.e., warm and wet for 184-241 CE versus cold and near/below average precipitation for 365-422 CE). The same issue is present in the 15th century, albeit in somewhat different form. The positive trend in temperature indicated by the oak isotopes (Fig. 4a) affected the PDSI towards drier conditions towards the end of the century.

The value of studying hydroclimatic indices that do *not* weight evapotranspiration heavily (unlike PDSI) is further supported by the stronger relationship between the documentary-based indices presented here and the precipitation (Dobrovolný et al., 2018) and water balance (Torbenson et al., 2023) reconstructions (Fig. 5).

Line 318: "they found no reflection in documentary sources" - Yes, other studies e.g. Harvey & Macdonald, 2021 have identified this truncation.

RESPONSE: We added here a new reference as follows: "... they found no reflection in documentary sources (*cf.* Harvey-Fishenden and Macdonald, 2021)."

New reference:

Harvey-Fishenden, A. and Macdonald, N.: Evaluating the utility of qualitative personal diaries in precipitation reconstruction in the eighteenth and nineteenth centuries, Clim. Past, 17, 133–149, https://doi.org/10.5194/cp-17-133-2021, 2021.

Lines 370-371: this critical reflection of the model here is key - it has low constraints/inputs - higher uncertainty.

RESPONSE: This note concerns the quality of the ModE-RA reanalysis data. We are aware of higher uncertainty in ModE-RA data for the 15th century. Please see our response to Referee #2 (Point 3, last paragraph) for more details.

**Line 421:**

RESPONSE: Corrected as: "indices correspond well with"

End of the last paragraph before Discussion (line 437): The unknown here is the role of dry winters and the potential to exacerbate the impacts of 'warm-dry summers' as environments start the summer months in 'water' deficit, thereby potentially exacerbating the impacts felt (which could be documented in the sources) by communities, worthy of a comment? RESPONSE: We agree with the referee's opinion that the perception of the character of individual seasons (in this case of dry summers) can be significantly influenced by the weather conditions in previous seasons in the case of documentary sources. Unfortunately, the density of the Czech documentary evidence in the 15th century especially about wet/dry character of seasons (precipitation indices) is too low to prove this influence as can be seen from Fig. 2. Moreover, Czech indices indicate rather the persistent nature of wet winters and wet summers (e.g., 1433 or 1496) than dry winters and summers.

Line 504: "occurred in 1477 (Toohey and Sigl, 2017)" - Veidivotn eruption in Iceland RESPONSE: The corresponding sentence was complemented as follows: "Conversely, the only major eruption of the 15th century identified in the Greenland ice cores occurred in 1477 (Toohey and Sigl, 2017), associated with the Veiðivötn–Bárðarbunga volcanic system (Abbott et al., 2021)."

---

## Author Comment (AC5)

**Comment on Macdonald on Brázdil et al. in review by Christian Pfister**

RESPONSE: We would like to thank Christian Pfister for his comments to the review of Neil Macdonald. Our responses to Neil Macdonald corrections and suggestions are detailed in responses to his review.

Line 2 agreed – correctly line 10

Line 15-16 agreed

Line 19 I think that information on winter is indeed unique

Lines 52 and 55 corresponding quotations should be provided

Line 64 agreed

Line 65 agreed

Line 77 are these details really needed?

Line 84 Although religious wars had left the Czech Lands totally devastated by the end of the 15th century,

Line 90 agreed

Line 127 keep the sentence – OK

Line 167 add reference Dobrovolny et al 2010

Line 231 agreed

Line 248 agreed

Line 318 agreed

Line 421 add severity

Line 437 this is an interesting point. Perhaps the Central European indices by Pfister would be of use

Line 504 agreed